# Cytokinin functions as an asymmetric and anti-gravitropic signal in lateral roots

Sascha Waidmann [1], Michel Ruiz Rosquete[1], Maria Schöller[1], Elizabeth Sarkel[1], Heike Lindner[2], Therese LaRue[2,3], Ivan Petřík[4], Kai Dünser[1], Shanice Martopawiro[5], Rashmi Sasidharan [5], Ondrej Novak [4], Krzysztof Wabnik[6], José R. Dinneny[2,3] & Jürgen Kleine-Vehn [1]

Directional organ growth allows the plant root system to strategically cover its surroundings. Intercellular auxin transport is aligned with the gravity vector in the primary root tips, facilitating downward organ bending at the lower root flank. Here we show that cytokinin signaling functions as a lateral root specific anti-gravitropic component, promoting the radial distribution of the root system. We performed a genome-wide association study and reveal that signal peptide processing of Cytokinin Oxidase 2 (CKX2) affects its enzymatic activity and, thereby, determines the degradation of cytokinins in natural *Arabidopsis thaliana* accessions. Cytokinin signaling interferes with growth at the upper lateral root flank and thereby prevents downward bending. Our interdisciplinary approach proposes that two phytohormonal cues at opposite organ flanks counterbalance each other's negative impact on growth, suppressing organ growth towards gravity and allow for radial expansion of the root system.

---

[1] Department of Applied Genetics and Cell Biology, University of Natural Resources and Life Sciences, Vienna (BOKU), Muthgasse 18, 1190 Vienna, Austria. [2] Department of Biology, Stanford University, 260 Panama Street, Stanford, CA 94305, USA. [3] Department of Plant Biology, Carnegie Institution for Science, 260 Panama Street, Stanford, CA 94305, USA. [4] Laboratory of Growth Regulators, Centre of the Region Haná for Biotechnological and Agricultural Research, Faculty of Science of Palacký University and Institute of Experimental Botany of the Czech Academy of Sciences, Šlechtitelů 27, 78371 Olomouc, Czech Republic. [5] Plant Ecophysiology, Institute of Environmental Biology, Utrecht University, Padualaan 8, Utrecht 3584 CH, The Netherlands. [6] Centro de Biotecnología y Genómica de Plantas (Universidad Politécnica de Madrid - Instituto Nacional de Investigación y Tecnología Agraria y Alimentaria), Autopista M-40, Km 38-Pozuelo de Alarcón, 28223 Madrid, Spain. Correspondence and requests for materials should be addressed to J.K.-V. (email: juergen.kleine-vehn@boku.ac.at)

Root architectural traits define plant performance and yield[1]. The radial spreading of the root system depends on the directional growth of primary and secondary roots. The phytohormone auxin plays a central role in aligning root organ growth towards gravity[2]. In the root tip, columella cells perceive changes in gravity via statolith sedimentation[3]. The relative change in statolith positioning triggers a partial polarization of redundant PIN3, PIN4, and PIN7 auxin efflux carriers towards this side, leading to enhanced auxin transport along the gravity vector[4,5]. The asymmetric distribution of auxin eventually reduces cellular elongation rates at the lower root flank, which consequently leads to differential growth within the organ and bending towards gravity[6–8].

Lateral roots (LRs) substantially differ from main roots, establishing a distinct gravitropic set point angle (GSA)[9]. The divergent developmental programs of lateral and main (primary) roots allow the root system to strategically cover the surrounding substrate. In *Arabidopsis*, LRs emerge from the main root at a 90° angle (stage I LRs) and afterwards display maturation of gravity sensing cells, as well as the de novo formation of an elongation zone[7]. Transcription factors FOUR LIPS and MYB88 define PIN3 expression in columella cells of LRs[10] and its transient expression in columella cells temporally defines asymmetric auxin distribution as well as differential elongation rates in stage II LRs[7,11]. This developmental stage lasts about 8–9 h and is characterized by asymmetric growth towards gravity at a slower rate than in primary roots[7,12]. During this phase of development, the primary GSA of LRs is established. The subsequent repression of PIN3 in columella cells of stage III LRs coincides with symmetric elongation, maintaining this primary GSA[7]. Notably, the derepression of PIN3 and PIN4 in columella cells of older stage III LRs does not necessarily correlate with additional bending to gravity[8]. This finding illustrates that the primary GSA is developmentally maintained, determining an important root architectural trait. Moreover, a stage III LRs will return to its initial GSA if it is reoriented relative to the gravity vector[7,13,14]. Accordingly, the partial suppression of a full gravitropic response in recently emerged LRs is critical for establishing the primary growth direction of LRs, which importantly contributes to the root system architecture.

Despite the apparent importance of directional LR growth for radial exploration of the root system, the underlying suppressive mechanisms are largely unexplored. Using genetic, physiological, computational, biochemical, and cell biological approaches, we reveal that two opposing hormonal cues at the lower and upper lateral root flank counterbalance each other and set directional LR growth.

## Results

### Angular lateral root growth displays substantial natural variation.
To examine the natural diversity in radial root growth, we screened 210 sequenced *Arabidopsis* accessions (Supplementary Fig. 1a, Supplemental Data 1) and quantified their primary GSA of LRs. When grown in vitro on the surface of the growth medium, we observed extensive variation for the mean GSA values, detecting a deviation of about 40° between most extreme natural accessions (Fig. 1a).

Because the in vitro approach allowed only two-dimensional analysis of root growth, we further assessed angular growth of LRs in three dimensional and soil systems. For this purpose, we studied a subset (depicted by red and blue lines in Fig. 1a) of hyper-responsive and hypo-responsive accessions in greater detail (Fig. 1b, c Supplementary Data 2). To allow three-dimensional root expansion in vitro, we grew this subset of accessions in growth medium-filled cylinders[8] (Supplementary Fig. 1b). In

addition, we used the GLO-Roots system[15], which is a luciferase (LUC)-based imaging platform to visualize root systems in a soil-like environment (Supplementary Fig. 1c). Accordingly, we transformed the same subset of accessions with *pUBQ:LUC2o*[15], ubiquitously driving LUC expression. In the *Col-0* reference accession, about 60% of emerged LRs displayed a GSA between 51° and 70° in all three growth conditions (Fig. 1b–e, Supplementary Data 2). In all three systems, hypo-responsive and hyper-responsive accessions displayed a pronounced shift towards higher (71–90° and 91–110°) and lower (31–50° and 0–30°) angle categories, respectively (Fig. 1b–e, Supplementary Data 2). This suggests that our two-dimensional, in vitro screen was highly suitable to identify natural accessions with diverging GSA values of their root systems.

### GWAS reveals a link between cytokinin and angular growth of LRs.
Next, we sought to identify molecular players involved in the LR trait of our interest. To achieve this, we used our quantitative data on primary GSA of LRs and conducted a genome-wide association study (GWAS)[16]. We identified several chromosomal regions, displaying associations with our trait (Fig. 2a). A prominent peak at chromosome 2 drew our attention to a thymine (T)/guanine (G) single-nucleotide polymorphism (SNP) located in the *CYTOKININ OXIDASE 2* (*CKX2*) gene (position 8,447,233) (Fig. 2b). Importantly, the minor G allele, showing a frequency of 19.5% in all sequenced and 32.7% in our set of accessions, was associated with increased GSA values, reflecting more perpendicular LR growth to gravity (Fig. 2c). Notably, linkage disequilibrium analysis showed that adjacent SNPs display pronounced non-random association with our SNP of interest (Supplementary Fig. 2), suggesting that the *CKX2* gene could be linked to variations in angular growth of LRs.

CKX enzymes are responsible for the irreversible degradation of cytokinins (CKs) via the oxidative cleavage of their side chain[17]. iP-type CKs are the preferred substrate of CKXs[18] and were as expected increased in *ckx2-1* (*Col-0* background) mutant roots (Supplementary Fig. 3a). On the other hand, other types of CKs were downregulated in the *ckx2-1* mutant background (Supplementary Fig. 3a–e), presumably underlying a compensation mechanism. This measurement reveals that CK metabolism is indeed affected in *ckx2-1* mutant roots (Supplementary Fig. 3a–e). To test whether active CKs may modulate angular growth of LRs, we initially transferred 7-day-old seedlings of the reference accession *Col-0* to medium supplemented with CKs. We observed a strong concentration-dependent increase in GSA values of LRs emerging in presence of active CKs, such as 6-Benzylaminopurin (BAP) (Fig. 2d, Supplementary Data 2), *trans*-zeatin (tZ) and isopentenyladenine (iP) (Supplementary Fig. 3f, g, Supplementary Data 2). Conversely, CK receptor double mutant combinations showed a relatively mild, but statistically significant decrease in GSA of LRs (Fig. 2e, Supplementary Data 2), proposing functional redundancy among the cytokinin receptors.

This set of data suggests that cytokinin signaling interferes with downward bending of emerged LRs. Notably, emerging LRs of winter oilseed rape also displayed reduced bending of LRs when treated with BAP (Supplementary Fig. 3h, Supplementary Data 2), suggesting that the effect of CK on directional LR growth is likely to be conserved.

To further assess the importance of CKX2 in GSA establishment, we disrupted CKX activity in the reference accession *Col-0*. Treatments with the CKX inhibitor INCYDE[19] phenocopied the *ckx2-1* loss-of-function mutant, both displaying more horizontal LRs when compared to its respective controls (Fig. 2f, g, Supplementary Data 2). On the other hand, *CKX2* overexpressing (OX) plants showed accelerated bending of LRs, phenocopying

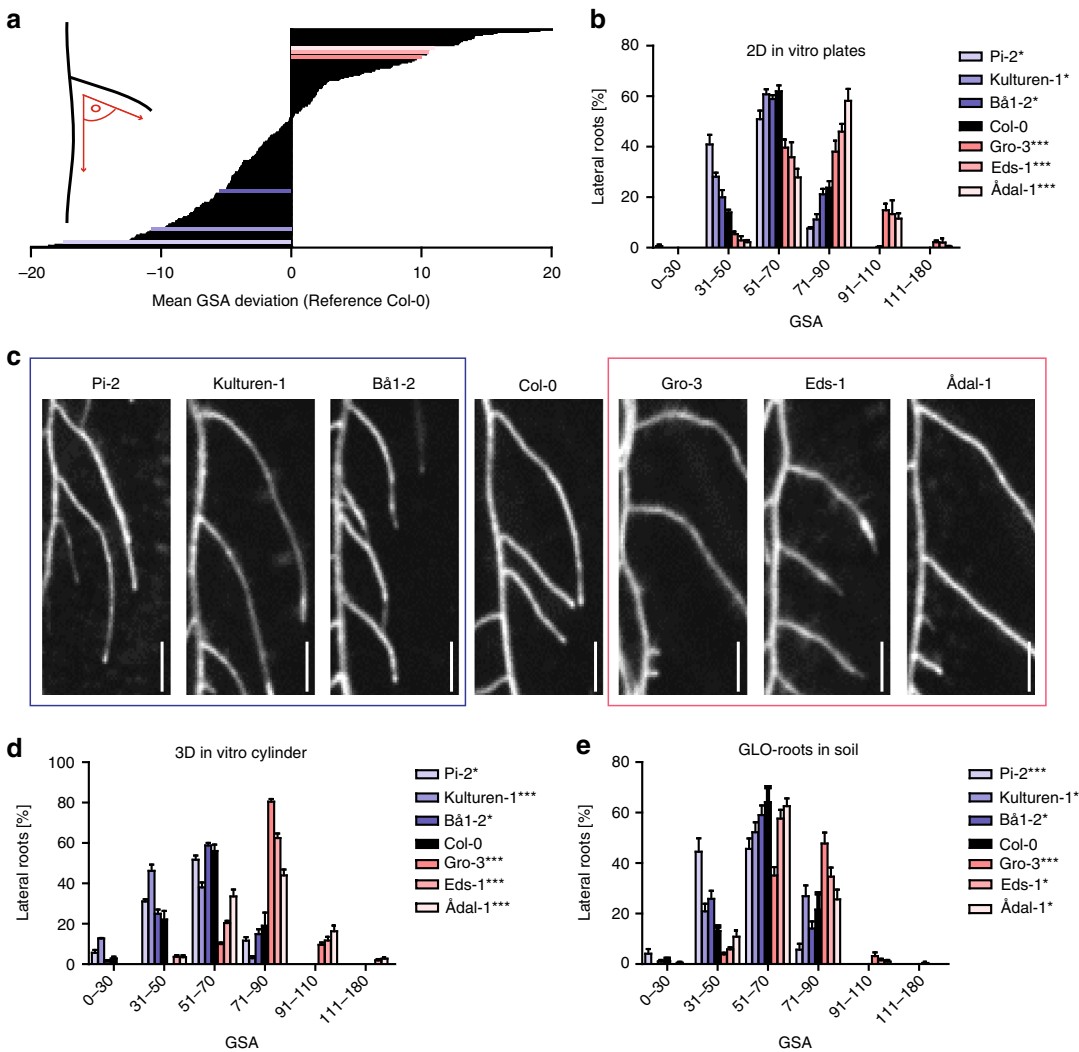

**Fig. 1** Natural variation of the primary GSA of lateral roots in *Arabidopsis thaliana*. **a** Mean gravitropic set point angle (GSA) values are normalized to reference accession *Col-0*. Three hyper-responsive (blue colors) and hypo-responsive (red colors) accessions were selected for further analysis. **b** GSA distributions of hyper-responsive and hypo-responsive accessions grown on 2D agar plates. $n = 5$ plates (16 seedlings with 30–120 LRs per plate). **c** Representative images of hyper-responsive and hypo-responsive accessions grown on 2D agar plates. Scale bars, 20 mm. **d** GSA distribution of hyper-responsive and hypo-responsive accessions grown in 3D agar cylinders. $n = 5$ cylinders (25–120 LRs per cylinder). **e** GSA distribution of hyper-responsive and hypo-responsive accessions grown in soil. $n = 5$–10 plants (25–75 LRs per plant). **b**, **d**, **e** Kolmogorov–Smirnov test *P*-values: *$P < 0.05$, **$P < 0.01$, ***$P < 0.001$ (compared to *Col-0*). Mean ± SEM. Experiments were repeated at least three times

the CK receptor mutants (Fig. 2g, Supplementary Data 2). This set of data suggests that CK signaling defines directional lateral root growth by reducing LR bending after emergence.

**Cytokinin response factors define angular growth of LRs**. Our data indicates that CK signaling impacts the angular growth of emerged LRs. Therefore, we assessed whether CK-dependent transcription factors indeed have an impact on LR growth in the reference accession *Col-0*. It has been previously shown that CK signaling initiates transcriptional changes via the *Arabidopsis* response regulators (ARRs)[20–22] and the cytokinin response factors (CRFs)[23,24]. According to available LR organ-specific microarray data[25], type-B ARRs, such as ARR10 and ARR12, and type-A ARRs, such as *ARR3* and *ARR4* (Supplementary Fig. 4a, b) were strongly upregulated in mature LRs. However, we did not detect any expression of *ARR3, ARR4, ARR10,* and *ARR12* in young stage II LRs, using respective promoter GUS reporter lines (*pARR3::GUS, pARR4::GUS, pARR10::GUS, pARR12::GUS;*

Supplementary Fig. 4d)). In agreement, angular growth of LRs was not altered in *arr3, arr4, arr10,* and *arr12* single mutants (Supplementary Fig. 4e, Supplementary Data 2). While this approach was not successful in identifying the responsible ARRs in young LRs, we in contrast could confirm expression of *CRF2* and *CRF3* ([25], Supplementary Fig. 4c) in the early stages of LR development (Fig. 3a and Supplementary Fig. 4f). *pCRF2::GFP-GUS* was ubiquitously expressed in young LRs, while *pCRF3::GFP-GUS* was preferentially expressed in cortical and epidermal cell files (Fig. 3a and Supplementary Fig. 4f). Notably, compared to emerged laterals, the main root displayed much weaker *CRF2* and *CRF3* expression (Supplementary Fig. 4g), suggesting that *CRF2* and *CRF3* expression is particularly high in young LRs.

In agreement with *CRF2* and *CRF3* expressions in emerged LRs, loss-of-function alleles of *crf2* and *crf3* displayed enhanced bending of LRs (Fig. 3b, Supplementary Fig. 4h, Supplementary Data 2). Conversely, we found that ubiquitous overexpression of either CRF2 or CRF3 led to more horizontal LRs (Fig. 3b, Supplementary Data 2). When we transferred *crf2* and *crf3* single

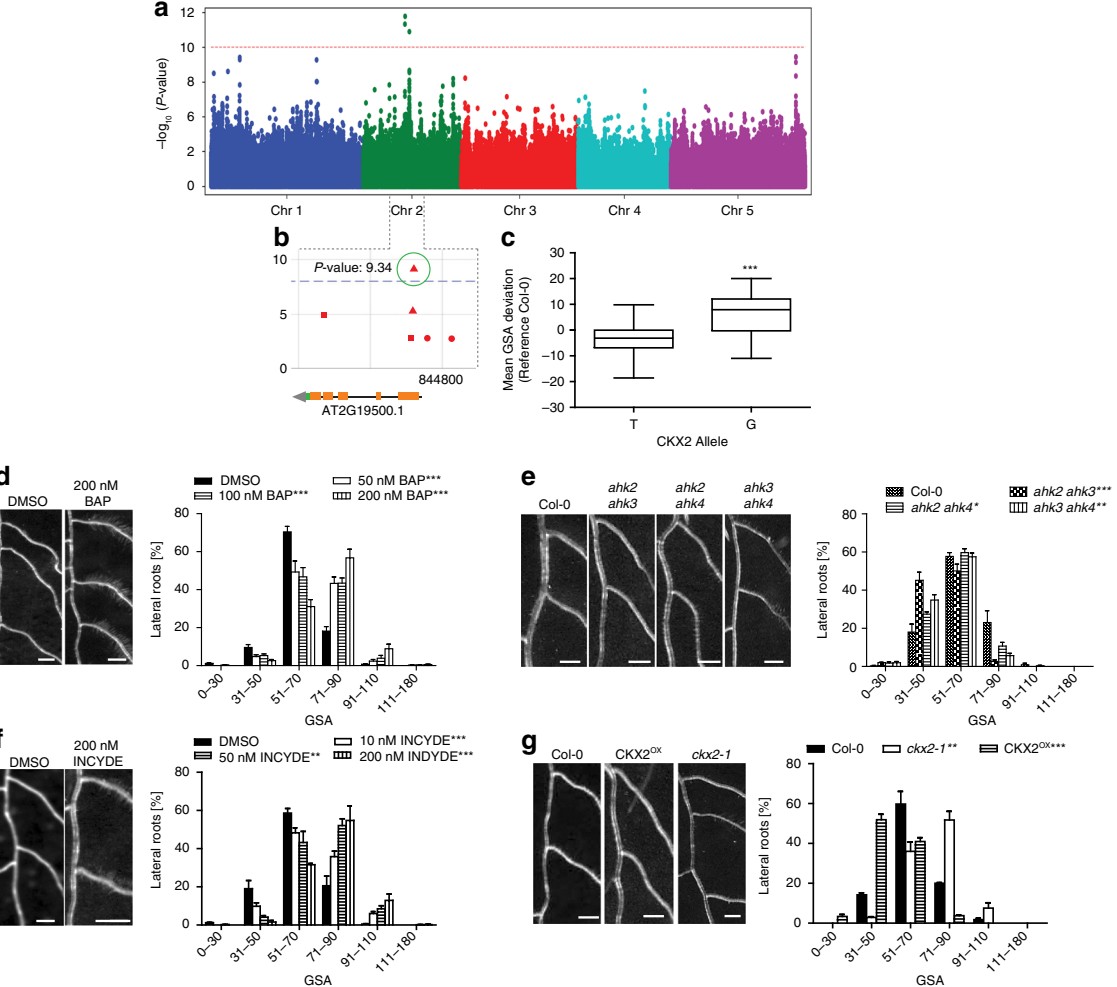

**Fig. 2** Genome-wide association study (GWAS) on gravitropic set point angle (GSA). **a** Manhattan plot of GWAS results. The dotted horizontal line indicates a significance level of 0.1 after Bonferroni correction for multiple testing. **b** Magnification of the peak region on chromosome 2. A highly significant SNP was located at position 8,447,233 in the coding region of *CKX2*. **c** Mean GSA of T and G alleles of CKX2. Horizontal lines show the medians; box limits indicate the 25th and 75th percentiles; whiskers extend to the min and max values. Student's *t*-test *P*-value: ***$P < 0.001$. **d**–**g** Representative images and GSA distributions of untreated and 6-Benzylaminopurin (BAP)-treated Col-0 wild type **d**, Col-0 wild type, *ahk2 ahk3*, *ahk2 ahk4* and *ahk3 ahk4* **e**, untreated and INCYDE-treated Col-0 wild type **f**, Col-0 wild type, *ckx2-1* and CKX2^OX seedlings **g**. Kolmogorov–Smirnov test *P*-values: *$P < 0.05$, **$P < 0.01$, ***$P < 0.001$ (compared to DMSO solvent or Col-0 wild type control). Mean ± SEM, $n = 5$ plates (16 seedlings with 65–160 LRs per plate). Scale bars, 2 mm. **d**–**g** Experiments were repeated at least three times

mutants to medium supplemented with CK, we observed partial resistance to CK in both lines (Supplementary Fig. 4i, Supplementary Data 2). This set of data confirms that cytokinin signaling, utilizing transcription factors, such as CRF2 and CRF3, regulates angular growth of LRs.

**Cytokinin integrates environmental cues into angular growth of LRs.** Our data supports a role for cytokinin signaling in modulating angular growth of LRs. To investigate whether cytokinin modulates angular LR growth in response to environmental cues, we examined whether the primary GSA of *Arabidopsis* accessions is linked to geographic origins. Intriguingly, accessions with the largest GSA values predominantly originated in Nordic (above 58°N) regions (Fig. 4a). In addition, the above described minor G allele of *CKX2*, phenocopying the *ckx2-1* loss of function (in *Col-0*), was notably the most prevalent allele in the north of Sweden (Fig. 4a, b). Previous work showed that *Arabidopsis* accessions in the north of Sweden are fully vernalized before snow fall, but would sit out winter and only flower in the

next spring[26]. In fact, the respective habitat in the north of Sweden is most of the year covered with snow (Supplementary Fig. 5a), indicating that these fully-grown leaf rosettes and root systems withstand long-term snow coverage. Snowpack insulation capacity can protect these plants from extreme temperatures, but may also restrict soil–atmosphere gas exchange, eventually leading to hypoxia in the soil[27]. Additionally, rapid snowmelt in spring can lead to temporary soil flooding, which depletes soil oxygen and may restrict the amount of oxygen reaching the root tissues. Intriguingly, endogenous hypoxia in roots may repress lateral root primordia development[28] and hypoxic conditions induce an organ-bending response in the primary root as a possible adaptive avoidance response[29]. Therefore, we asked whether hypoxia conditions also modulate the bending of LRs. In contrast to the primary root response, we observed that hypoxic stress reduced bending in emerged LRs (Fig. 4c, Supplementary Data 2), demonstrating distinct pathways to regulate root bending in primary and secondary roots. Interestingly, hypoxia stress for 4 h was sufficient to increase GSA of subsequently emerged LRs in *Col-0* (Fig. 4c, Supplementary Data 2), mimicking the *ckx2-1*

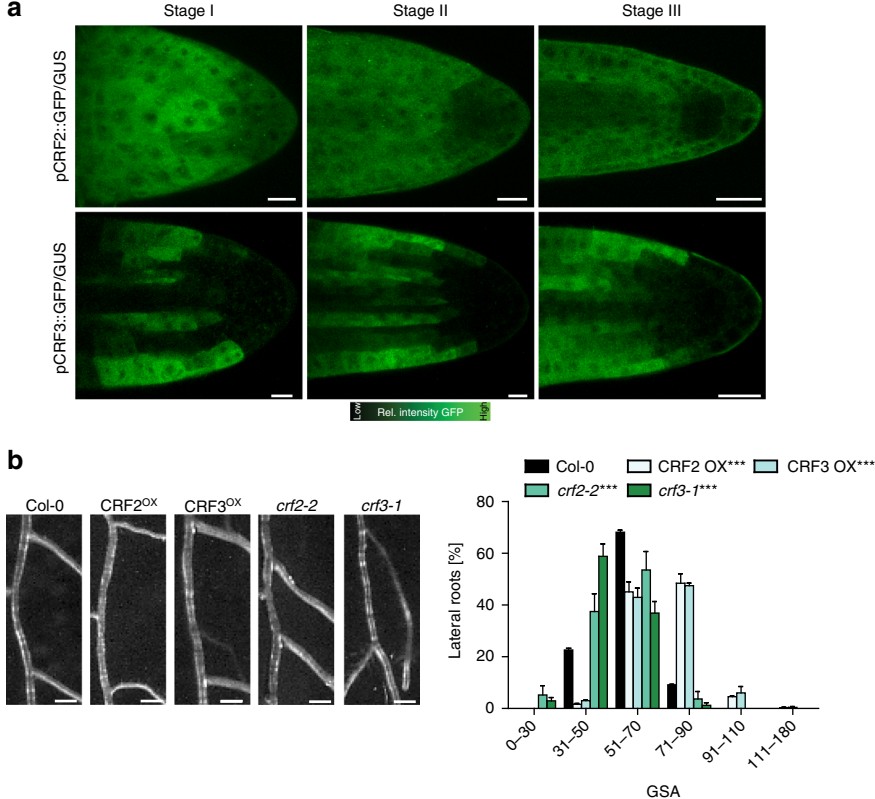

**Fig. 3** Characterization of cytokinin response factors (CRFs) in lateral roots. **a** Representative images of pCRF2::GFP/GUS and pCRF3::GFP/GUS in stage I–III LRs. Scale bar, 25 μm. **b** Representative images and GSA distribution of *Col-0* wild type, *crf* mutants, and CRF^OX lines. Kolmogorov–Smirnov test *P*-values: ***P < 0.001 (compared to DMSO or *Col-0*). Mean ± SEM, *n* = 5 plates (16 seedlings with 50–120 LRs per plate). Scale bars, 2 mm. **a**, **b** Experiments were repeated at least three times

loss of function phenotype. Furthermore, hypoxic stress did not further increase GSA in the *ckx2-1* mutant (Fig. 4c, Supplementary Data 2), proposing that CK metabolism could mediate hypoxia-dependent repression of LR bending. Furthermore, the LRs of *ahk2 ahk4* CK receptor mutants were insensitive to the hypoxia-induced repression of LR bending (Fig. 4d, Supplementary Data 2). Accordingly, we conclude that CK signaling integrates environmental signals, such as hypoxia, into GSA establishment of emerged LRs.

**Single base-pair variation in *CKX2* impacts on its in planta activity**. Our data proposes that variation in *CKX2* is linked to the control of radial root system expansion in natural *Arabidopsis* accessions. The previously mentioned G allele of *CKX2* is associated with higher GSA values (Fig. 2c), which phenocopies the loss of *CKX2* function or increase in CK levels in the reference accession *Col-0* (Fig. 2d, g, Supplementary Data 2). Accordingly, we next assessed whether the identified SNP affects the activity of CKX2. The underlying T to G mutation alters the first amino acid in the mature enzyme from an isoleucine (I) to a methionine (M). This mutation is situated just after the predicted cleavage site of a signal peptide (SP). The SP allows CKX2 to be inserted into the endoplasmic reticulum and to be subsequently secreted into the extracellular space (apoplast)[17]. To assess potential trafficking or processing defects caused by the amino acid change[30], we generated a ratiometric CKX2 reporter by fusing the green fluorescent protein (GFP) and mScarlet to the N-terminal and C-terminal ends of CKX2, respectively. Fluorescent mScarlet signal of the non-mutated CKX2^I readily accumulated in the apoplast, suggesting that the fluorescent tags do not abolish processing

and/or secretion of CKX2-mScarlet (Fig. 5a and Supplementary Fig. 5b). Ratiometric imaging of GFP and mScarlet revealed a higher degree of co-localization for mutated version CKX2^M, suggesting reduced processing and/or secretion of CKX2^M when compared to CKX2^I (Fig. 5b). To visualize the effect of the T to G mutation on SP cleavage, we N-terminally tagged CKX2 with GFP and subsequently expressed GFP^SPCKX2^I and its respective mutated version GFP^SPCKX2^M in tobacco. Western blot analysis revealed a decreased cleavage of GFP^SPCKX2^M when compared to GFP^SPCKX2^I (Fig. 5c). Even though we cannot eliminate the possibility that N-terminal GFP may interfere with normal SP-processing rates, the relative differences between the two assessed alleles suggests that the identified SNP reduces the SP cleavage in CKX2.

The SP processing is an important determinant of the mature protein and, hence, we examined the enzymatic CKX2 activity in the presence and absence of the signal peptide. We expressed full length ^SPCKX2^I and ^SPCKX2^M as well as the SP-lacking counterparts ^-SPCKX2^I and ^-SPCKX2^M in *Escherichia coli* and measured their ability to oxidize CKs. Both SP-lacking forms ^-SPCKX2^I and ^-SPCKX2^M showed a 10-fold higher activity compared to the SP containing versions (Fig. 5d). This in vitro data suggests that SP processing is required to ensure full enzymatic activity of CKX2.

Next, to assess whether the T to G mutation also affects CKX2 activity in planta, we expressed full length *pCKX2::CKX2^I* and *pCKX2::CKX2^M* encoding versions in the *ckx2-1* mutant background. As expected, the wild-type (Col-0) *CKX2^I* was able to fully complement the *ckx2-1* mutant phenotype (Fig. 5e, Supplementary Fig. 5c, Supplementary Data 2). In contrast,

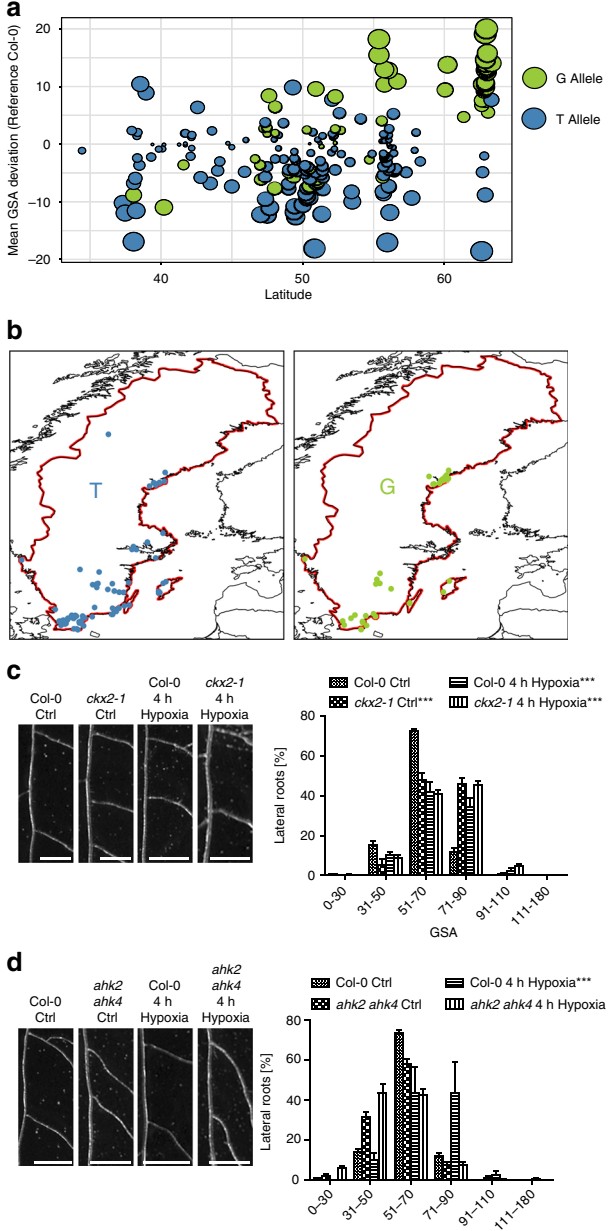

**Fig. 4** Cytokinin signaling integrates environmental signals into angular lateral root growth. **a** Comparison of the mean GSA distribution and its geographical (latitude) distribution of the phenotyped accessions. T and G allele of CKX2 are depicted in blue and green, respectively. **b** Relative geographical distribution of the T and G allele of CKX2 in all sequenced Swedish *Arabidopsis* accessions. The distribution of accessions is visualized by R package "rworldmap". **c**, **d** Representative images and GSA distributions of **c** Col-0 wild-type and *ckx2-1* or **d** Col-0 wild type and *ahk2 ahk4* with and without hypoxia treatment for 4 h. Scale bars, 2 mm. Kolmogorov–Smirnov test *P*-values: ***$P < 0.001$ (compared to DMSO solvent or Col-0 wild type control). Mean ± SEM, $n = 4$ plates (10 seedlings with 60–120 LRs per plate). Experiments were repeated at least three times

the mutated $CKX2^M$ version was not able to reverse the reduced LR bending of *ckx2-1* mutants (Fig. 5e and Supplementary Fig. 5c). Overall, our data suggests that the T to G mutation found in natural accessions renders *CKX2* to be largely non-functional in planta by disrupting its secretion and/or SP processing.

Thus, we conclude that variation in SP processing of CKX2 contributes to the natural variation of CK-dependent angular LR growth in *Arabidopsis*.

**CKX2 does not detectably interfere with auxin signaling in emerged LRs**. Next, we investigated the cellular mechanism by which CKX2 activity modulates the primary GSA of LRs. We first inspected the spatial expression of *CKX2* to identify cells in which *CKX2* may directly regulate angular growth in LRs. pCKX2::CKX2-mTurquoise was weakly expressed in the tip of stage I LRs but showed increased expression in stage II and III LRs (Fig. 6a). We confirmed that endogenous *CKX2* transcripts are strongly up-regulated in stage II and III LRs by examining expression in excised LR tissue using qPCR (Fig. 6b). Notably, *pCKX2::CKX2-mTurquoise* was not readily detectable in the primary root tip (Fig. 6b and Supplementary Fig. 6a), proposing that CKX2 might preferentially act in secondary root organs. In agreement, gravitropic response of *ckx2-1* mutant main roots were largely not distinguishable from wild type roots (Supplementary Fig. 6b).

We next aimed to investigate how deviations in CKX2-dependent modulation of CK in LRs may modulate their directional growth. CKs signaling impairs PIN-dependent auxin transport in main roots as well as in LR primordia[31]. We therefore assessed whether CKX2 activity regulates auxin transport in emerged LRs. Because PIN3 is the main regulator of asymmetric auxin redistribution in columella cells of emerged LR[7], we initially assessed whether the *ckx2-1* mutant shows defective abundance or localization of functional pPIN3::PIN3-GFP in columella cells. At the time of GSA establishment (stage II LRs), PIN3-GFP abundance and asymmetry were not detectably altered in *ckx2-1* mutants when compared to wild type (Fig. 6c and Supplementary Fig. 6c). Next, we used the auxin responsive promoter DR5 fused to GFP and assessed whether auxin signaling is affected in *ckx2-1* mutant LRs. In accordance with proper PIN3 localization, DR5 signal intensity in columella cells, and asymmetric signal in the flanks was similar in *ckx2-1* mutant and wild type stage II LRs (Fig. 6d and Supplementary Fig. 6d).

Overall, this set of data illustrates that auxin responses in gravitropic LR are not detectably altered by CKX2, suggesting that CKX2 modulates angular growth by an alternative, CK-dependent mechanism in emerged LRs.

**Emerged LR display asymmetric cytokinin signaling**. Our data indicates that CK regulates angular LR growth. To further assess the mechanism by which CK modulates GSA establishment in developing LRs, we visualized the spatial distribution of CK signaling, using the two-component signaling sensor (TCSn) transcriptionally fused to GFP (TCSn::GFP)[32]. We observed increased CK signaling on the upper side of stage II LRs, coinciding with gravitropic bending (Fig. 6e). This asymmetry declined in stage III LRs, which maintain the previously established GSA (Fig. 6e and Supplementary Fig. 6e). In agreement with the anticipated reduction in CK degradation, the magnitude of asymmetric CK signaling was increased in *cxk2-1* mutant LRs (Fig. 6f and Supplementary Fig. 6e). Conversely, asymmetric CK signaling was reduced in the CK receptor double mutant *ahk2 ahk4* (Fig. 6g and Supplementary Fig. 6f). These data propose that the increased magnitude of asymmetry in CK signaling across the root tip correlates with reduced LR bending towards gravity.

To determine whether asymmetric CK signaling regulates bending specifically in LRs, we examined the distribution of CK signaling in primary roots responding to gravity. Importantly, we did not observe asymmetric CK signaling in unstimulated or gravity-stimulated primary roots (Supplementary Fig. 6g–h).

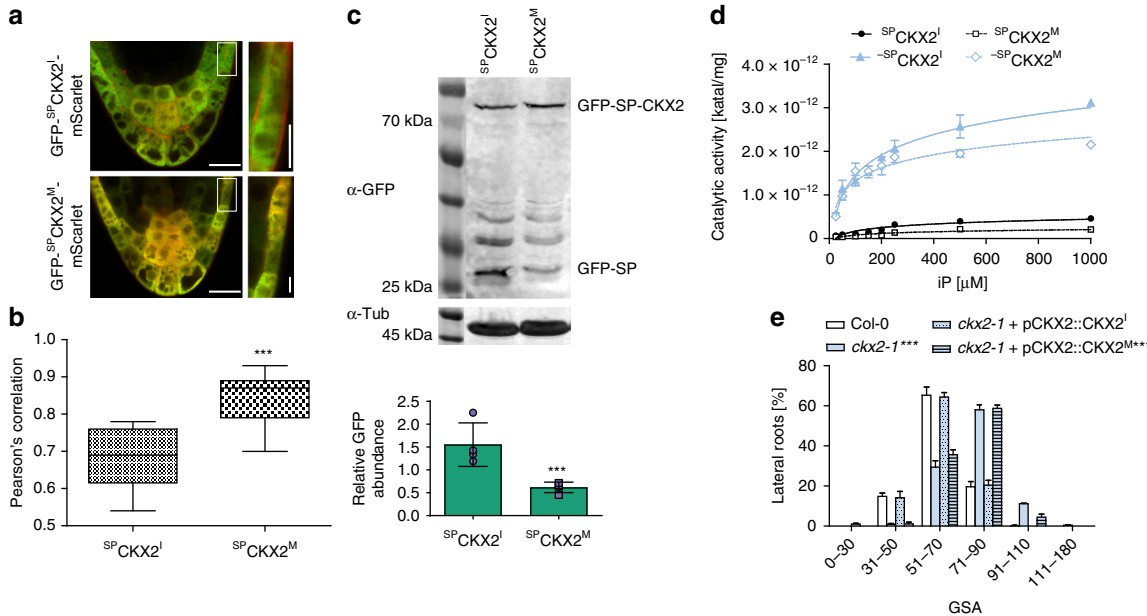

**Fig. 5** Signal peptide processing is required for CKX2 activity. **a** Localization of GFP-$^{SP}$CKX2$^I$-mScarlet and GFP-$^{SP}$CKX2$^M$-mScarlet in stage II LRs. Scale bar, 25 and 10 μm, respectively. **b** Quantification of the co-localization of GFP and mScarlet signal using Pearson's correlation. Horizontal lines show the medians; box limits indicate the 25th and 75th percentiles; whiskers extend to the min and max values. Student's t-test P-Value: ***P < 0.001, n = 10–12 individual LRs. **c** Immunoblot analysis and quantification of $^{SP}$CKX2$^I$ and $^{SP}$CKX2$^M$ expressed in N. benthamiana leaves using anti-GFP antibody. Anti-tubulin antibody was used as loading control. The signal of GFP-SP was quantified and normalized to tubulin. Student's t-test P-Value: ***P < 0.001. Mean ± SEM, n = 4 biological replicates. **d** Saturation curves of isopentenyladenine (iP) degradation by CKX2. Reactions were performed at pH 7.4 in McIlvaine buffer with 0.5 mM DCIP as electron acceptor (black filled circle $^{SP}$CKX2$^I$, white square $^{SP}$CKX2$^M$, black filled triangle $^{SP}$CKX2$^I$, white diamond $^{-SP}$CKX2$^M$). Mean ± SEM, n = 8. **e** GSA distributions of ckx2-1 was complemented by pCKX2::CKX2I, but not by pCKX2::CKX2M. Representative lines are shown. Kolmogorov–Smirnov test P-value: ***P < 0.001 (compared to Col-0). Mean ± SEM, n = 5 plates (16 seedlings with 65–160 LRs per plate). **a**–**e** Experiments were repeated at least three times

Accordingly, we conclude that TCSn-based asymmetric CK signaling is specific to LRs and thus contributes to the distinct establishment of primary GSA in LRs. Previous work proposed a hypothetical gravitropic offset component at the upper flank of LRs. This envisioned component was presumably sensitive to the inhibition of auxin transport[14]. To assess if auxin transport similarly modulates the asymmetry of CK signaling in emerged LRs, we treated seedlings with the auxin transport inhibitor 1-N-naphthylphthalamic acid (NPA). Pharmacological interference with auxin transport indeed markedly decreased asymmetric CK signaling in stage II LRs, when compared to the DMSO solvent control (Fig. 6h and Supplementary Fig. 6i), suggesting that auxin transport indeed impacts asymmetric CK signaling in emerged LRs.

In summary, our data suggests that asymmetric CK signaling at the upper flank of LRs functions as an anti-gravitropic component in emerged LRs to promote radial root growth.

**CKX2 activity determines cellular elongation in emerged LRs.** Light sheet-based live cell imaging showed that cells on the upper and lower flanks of emerged LRs show differential elongation for about 8–9 h[7]. During this developmental stage II, the cellular elongation rates at the upper epidermal layers is three-fold-increased compared to the lower flank (15 μm/h versus 5 μm/h)[7]. To test if this difference can account for the primary GSA establishment, we used these quantitative growth parameters to construct a dynamic computational model of LR bending (Fig. 7a–d and Supplementary Fig. 7a–f). This model incorporates cellular mechanics to simulate cell elongation using stretchable strings as a manifestation of the cell wall elasticity and internal

turgor pressure in the cell (see the "Methods" section). The anisotropic growth is simulated by extending the resting length of the string to account for three-fold differences in the growth rates between upper and lower flanks. The resulting model predicts that the incorporation of measured elongation rates on the upper LR flank is able to realistically recapitulate LR bending angle of wild type plants, reaching an angle of about 62–63° within 8–9 h (Fig. 7b, Supplementary Fig. 7a, b).

Next, we experimentally assessed whether the loss of CKX2 or CK application interferes with cell elongation in stage II LRs. Wild type seedlings showed asymmetric elongation (longer cell length at the upper compared to the lower flank) in stage II LRs (Supplementary Fig. 8a, b). In agreement with reduced LR bending, the ckx2-1 loss-of-function mutant, as well as wild type plants treated with BAP showed abolished asymmetry in cellular elongation (Supplementary Fig. 8a, b). Our previous work revealed that differential elongation is a major factor controlling LR bending[7]. However, the loss of CKX2 reduced cell elongation at the upper root flank in average only by 10% (Fig. 7e). To evaluate whether the measured reduction in cell length can realize the observed quantitative changes in LR bending, we reduced cellular elongation similarly by 10% in our computational LR model. The model predicted that CKX2-dependent impact on cellular elongation mildly increases the predicted GSA of LRs (Fig. 7c, d). Thus, we raise some suspicion that the impact of CKX2 on cellular elongation fully explains the observed reduction of LR bending in ckx2-1 mutants.

**Cytokinin alters cell division rates and defines angular growth of LRs.** In primary roots, CK reduces not only cellular

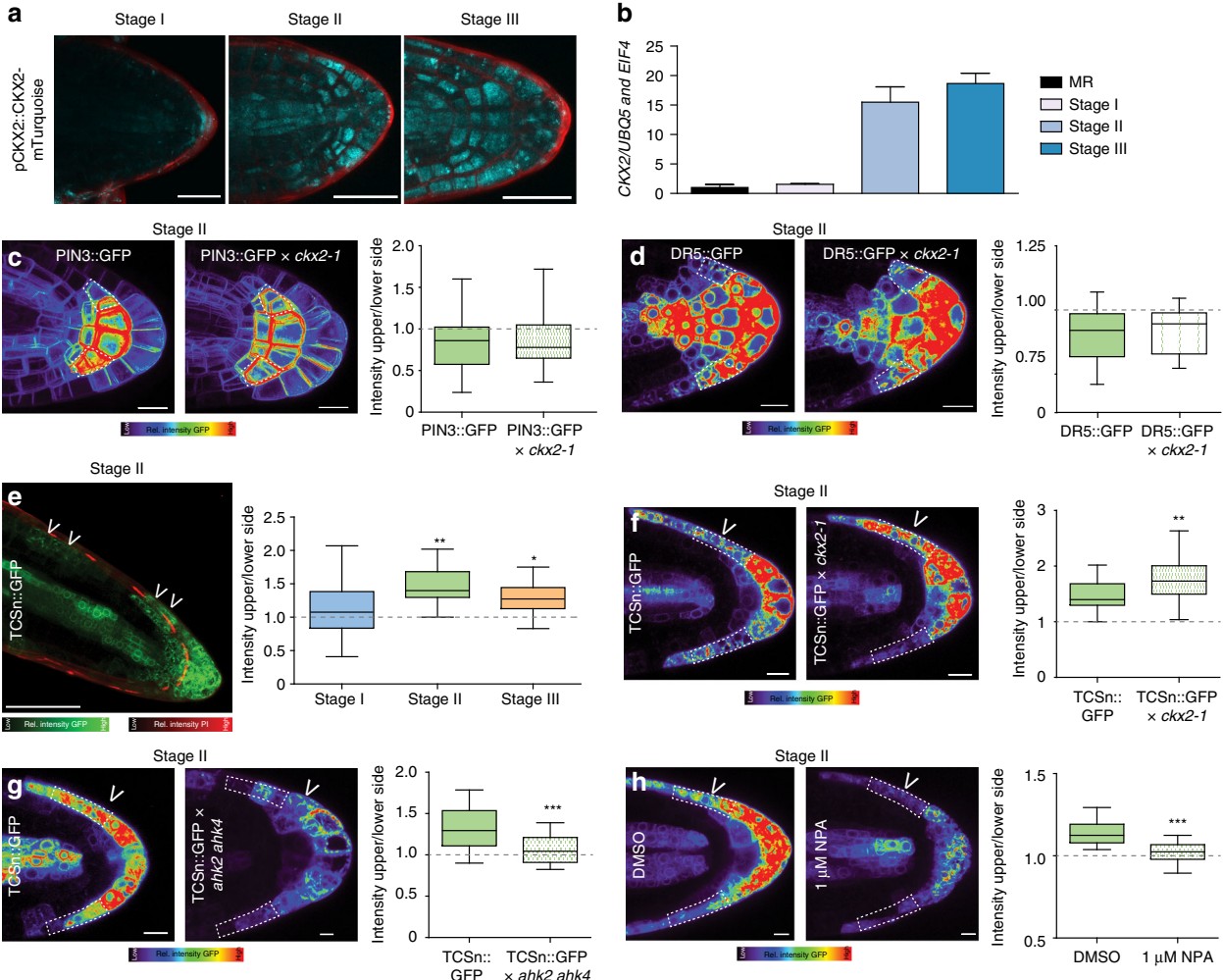

**Fig. 6** CKX2 modulates asymmetric cytokinin signaling in emerged lateral roots. **a** Representative images of pCKX2::CKX2-mTurquoise in stages I–III LRs. Propidium Iodide (PI) was used for counterstaining. Scale bar, 25 μm. **b** qPCR analysis detecting the levels of *CKX2* transcript in the root tip and LRs stages I–III normalized against *UBQ5* and *EIF4*. Bars represent means ± SD, *n* = 3. **c, d** Representative images and signal quantification of stage II LRs of **c** pPIN3:: PIN3-GFP, and **d** DR5::GFP in *Col-0* wild type and *ckx2-1* mutant background. Horizontal lines show the medians; box limits indicate the 25th and 75th percentiles; whiskers extend to the min and max values, *n* = 10–15 individual LRs. Scale bars, 10 μm. **e** Representative image (stage II) and quantification of TCSn::GFP in stages I–III LRs. PI was used for counterstaining. Scale bar, 50 μm. **f–h** Representative images and quantification of stage II LRs of **f** TCSn::GFP in wild type and *ckx2-1*, **g** TCSn::GFP in wild type and *ahk2 ahk4* or **h** after treatment with DMSO or 1 μM NPA for 24 h. Scale bars, 10 μm. **e–h** One-way ANOVA *P*-values: *$P < 0.05$, **$P < 0.01$, ***$P < 0.001$. Horizontal lines show the medians; box limits indicate the 25th and 75th percentiles; whiskers extend to the min and max values, *n* = 15–30 individual LRs. **a–h** Experiments were repeated at least three times. White dotted lines outline lateral root cap cells (facing the columella cells) for quantification

elongation, but also cell proliferation by distinct mechanisms[33,34]. Moreover, our computational model predicts that the rate of LR bending could be restricted by the number of cells (Supplementary Fig. 7a, b). Thus, we tested if CK might also affect the meristem of LRs and used cell division marker pCycB1;1::GUS to assess the spatial impact of CK on cell proliferation. BAP and INCYDE treatment reduced the abundance of pCycB1;1::GUS at the upper flank of stage II LRs (Supplementary Fig. 8c, d). In agreement, *ckx2-1* mutants displayed a more pronounced and statistically significant asymmetry in meristematic cell numbers when compared to wild type stage II LRs, revealing slightly less and more cell divisions at the upper and lower LR flanks, respectively (Fig. 7f). This set of data suggests that CK signaling not only restricts cellular elongation, but also induces a slight asymmetry in cell proliferation in emerged LRs.

Notably, *CDKB1;1* and other cell cycle promoting genes are down-regulated in the *crf1,3,5,6* quadruple mutant[23]. Hence, we assumed that CRF-dependent control of the cell cycle may contribute to the CK-mediated establishment of GSA in emerged LRs. To reduce cell cycle progression, we used the dominant negative (DN) allele of CDKB1;1 and the *cdkb1;1 cdkb1;2* double mutant (Fig. 7g, Supplementary Data 2), as well as the cell cycle inhibitor Roscovitine (Supplementary Fig. 8e, Supplementary Data 2). Both genetic and pharmacological interference with the cell cycle interfered with the LR bending (Fig. 7g, Supplementary Data 2, Supplementary Fig. 8e), phenocopying *ckx2-1* mutants. Intriguingly, CDKB1;1^DN as well as *cdkb1;1 cdkb1;2* showed a meristematic asymmetry, displaying a higher number of meristematic cells in the lower flank as compared to the upper flank (Fig. 7h). This is somewhat reminiscent to the *ckx2-1* mutant phenotype and we

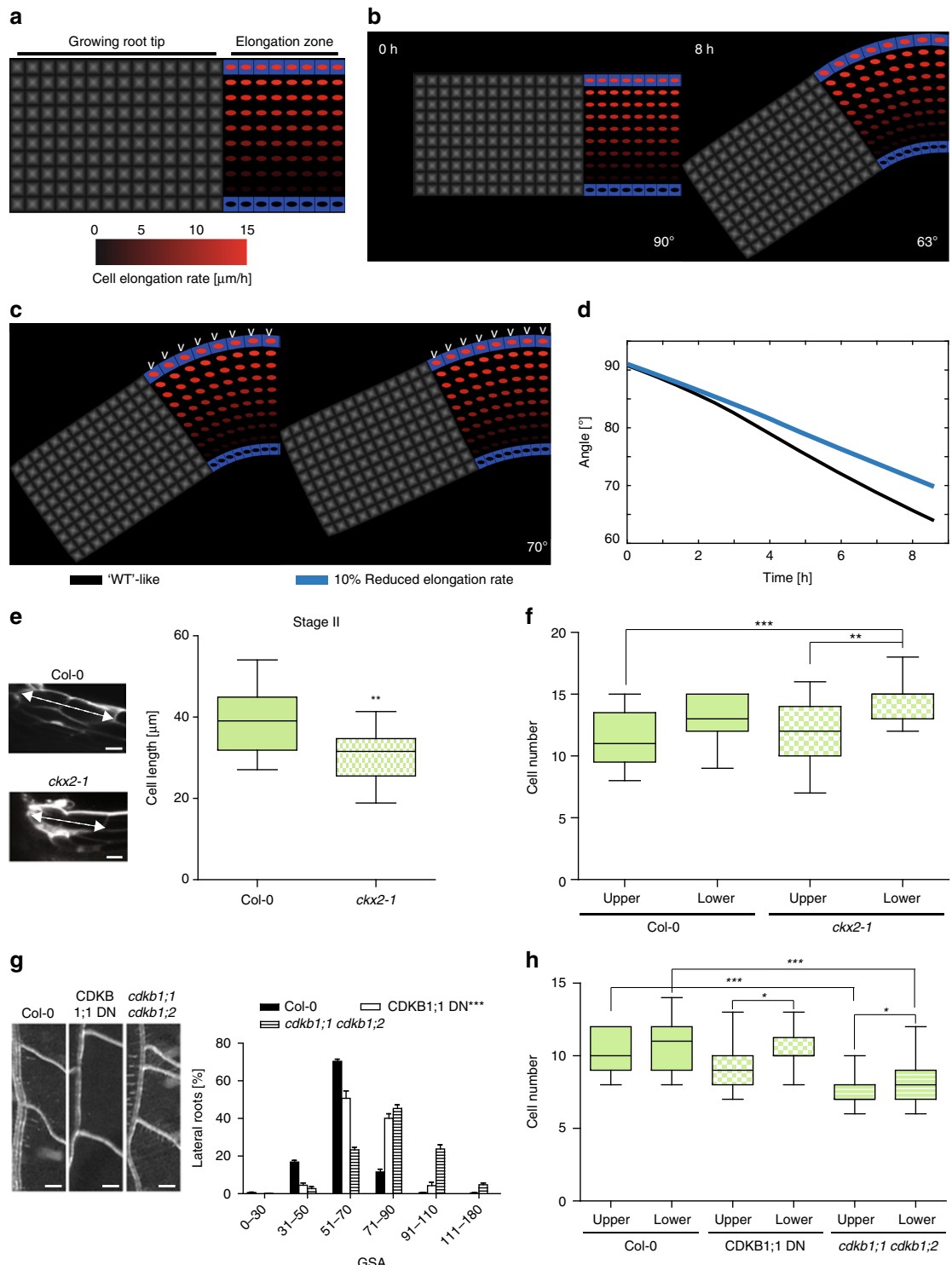

**Fig. 7** Cytokinin affects cell elongation and number in lateral roots. **a** Sketch shows a simplified geometry of a lateral root (LR). LR model consists of tip and elongation zones. The cell elongation rate (visualized as red spot inside the cell) linearly increases from lower flank towards the upper flank of the LR (up to three-fold) based on estimates derived from previous work[7]. Bottom panel, color coding bar for cell elongation rates. **b** Time-lapse model simulations (screenshots) which lead to the in vivo observed LR bending (63°) after ~8 h. **c** Left panel corresponds to **b**. Right panel, 10% decrease in elongation rate only on the upper root flank (white arrow heads). Each simulation represents LR status after 9 h of dynamic elongation. **d** Time evolution of set-point angle corresponding to different scenarios in **c**. **e** Representative image and quantification of first two elongated cells of lateral roots in stage II. **f** Quantification of the cell number in the upper and lower meristem of *Col-0* and *ckx2-1*. **g** Representative images and GSA distributions of *Col-0* wild-type, CDKB1;1 DN (dominant negative) and *cdkb1;1 cdkb1;2*. Kolmogorov–Smirnov test *P*-values: \*\*\**P* < 0.001 (compared to *Col-0*). Mean ± SEM, *n* = 5 plates (16 seedlings with 100–180 LRs per plate). Scale bars, 2 mm. **h** Quantification of the cell number in the upper and lower meristem of *Col-0*, CDKB1;1 DN and *cdkb1;1 ckdb1;2*. **e**, **f**, **h** One-way ANOVA *P*-values: \**P* < 0.05, \*\**P* < 0.01, \*\*\**P* < 0.001. Horizontal lines show the medians; box limits indicate the 25th and 75th percentiles; whiskers extend to the min and max values, *n* = 10–20 individual LRs. Scale bar, 10 μm. **e–h** Experiments were repeated at least three times

assume that the CK-dependent, asymmetric interference with the cell cycle is amplified in these cell cycle mutants when compared to wild type. In agreement with such a scenario, the gravity response kinetics of primary roots, which lack asymmetric TCSn-based CK signaling, were not affected by the CDKB1;1[DN] as well as *cdkb1;1 cdkb1;2* mutations (Supplementary Fig. 8f, g). This suggests that not only cellular elongation[7], but also the negative impact of CK on the cell cycle in stage II LRs is a particular determinant of directional LR growth.

Overall, this set of data suggests that CK modulates both differential cell elongation and cell proliferation to interfere with growth at the upper flank of LR, ultimately regulating angular LR growth and radial expansion of the root system.

## Discussion

Because root systems are hidden beneath the soil, the study and directed improvement of root architectural traits in crop breeding programs have been delayed. There is growing interest to alleviate the harmful effects of drought stress by modulating the primary GSA of LRs[1]. Despite the apparent importance of the root system depth, the molecular mechanisms regulating the direction of LR growth are poorly understood. Thus, understanding the molecular mechanisms establishing the primary angular growth in LRs could guide future engineering of plants to suit certain habitats. Anticipating that natural variation could provide valuable insights on how to sustainably engineer root systems, we focused on the primary growth direction of LR in natural *Arabidopsis* accessions.

We reveal that the primary growth direction of LR varies substantially within a population of natural *Arabidopsis* accessions. Primary LR angles of hypo-responsive or hyper-responsive accessions followed a similar trend regardless of whether they were grown in soil, two-dimensional or three-dimensional in vitro systems. We thus conclude that this approach is suitable to assess the genetic control of angular LR growth. Using a GWAS approach, we show that angular growth of LR is controlled by CKX2-dependent metabolism of the phytohormone CK. CKX2 contains a SP to enter the secretory pathway, which could be crucial for its impact on CK perception. However, the precise site of CK receptor activity (plasma membrane and/or endoplasmic reticulum) is still under debate[35]. We conclude that variation in an amino acid substitution after the predicted cleavage site impacts on SP processing of CKX2, which consequently obstructs CKX2 activity in planta. Our data suggest that the lack of SP processing abolishes the secretion and enzymatic activity of CKX2, thereby contributing to CK-dependent GSA trait variation in natural *Arabidopsis* accessions.

Nordic accessions preferentially express an inactive *CKX2* variant, which prompted us to investigate whether environmental cues further define the root system in a CK-dependent manner. We revealed that hypoxic conditions induce more horizontal LR growth through CK signaling. The increased frequency of an inactive *CKX2* allele in Nordic accessions suggests that the allele may have been selected for in these populations, promoting more horizontal root growth. It is an intriguing possibility that more horizontal, near surface roots may rectify gas exchange under hypoxia conditions, potentially alleviating the harmful effects of hypoxic stress in these Nordic, snow covered habitats. This assumption, however, awaits further experimental validation.

Our analysis suggests that primary and secondary roots have distinct responses to CK. While CK signaling abolishes PIN-dependent transport in main roots[31], we show that CKX2-dependent interference with endogenous CK levels does not affect PIN3 and auxin signaling in emerged LRs. Moreover, CK signaling is asymmetric in emerged lateral, but not primary roots, proposing a unique role of CK in regulating asymmetric growth responses in LRs. Also abscisic acid signaling displays distinct activities in main and lateral root organs, presumably allowing distinct organ growth rates in response to environmental stresses[36]. Thus, we propose that hormone signaling might be generally co-opted in primary and secondary roots to facilitate diverged growth responses to the environment.

The increase and decrease of CK levels slightly accelerate the rate of gravitropic bending in primary root, but the developmental importance of this effect remains uncertain[37]. In contrast, we show here that CK signaling plays a developmental role in establishing the primary GSA of LRs. Moreover, an increase and decrease of CK signaling correlate with reduced and enhanced down-ward bending of LRs, respectively. Mechanistically, we showed that CK signaling interferes with cellular elongation and proliferation in emerged LR to reduce LR organ bending towards gravity. These stage II LRs undergo a de novo formation of the elongation zone[7]. During this short developmental time window, the CK-dependent reduction in cell proliferation could have hence caused an immediate influence on the number of elongating cells. We detected only a mild asymmetry in cell numbers at the LR organ flanks of *ckx2* mutants, proposing that these LRs display only few cells less at the upper compared to the lower root flanks. Such a small impact could nevertheless compromise angular LR growth, because our computational model predicted that an asymmetric reduction in cell number at the upper root flank (Supplementary Fig. 7c–f) or gradually along the LR organ (Supplementary Fig. 9a–c) would induce some constraints, additionally limiting organ bending. However, this aspect awaits experimental validation, requiring detailed live cell imaging and possibly mechanical constraint measurements in LRs.

We illustrate that CKX2 contributes to the rate of asymmetric CK signaling, but *CKX2* expression did not show a pronounced asymmetry. Similarly, the CK response factors CRF2 and CRF3 are not asymmetrically expressed in emerged LR. We, hence, conclude that additional so far unknown factors play decisive roles in establishing asymmetric CK signaling across the LR tip. Our work proposes that an auxin transport mechanism promotes the asymmetry of CK signaling. Accordingly, auxin could generate an anti-gravitropic signal to interfere with its own gravitropic impact in LRs. Unlike auxin, the mechanisms of intercellular CK transport are poorly characterized[38]. One intriguing possibility is however that the asymmetric auxin signal could favor CK relocation towards the upper side of LRs, inducing differential CK signaling and growth repression on this side. However, it is also possible that differential CK signaling occurs at the level of signal integration and might be independent of differential distribution of CK. Future work will examine these possibilities to uncover the mechanism by which CKX2 is linked to differential CK activity across a stage II LR.

At the transition to stage III, *PIN3* expression in columella cells transiently decreases[7,11]. This temporal absence of PIN proteins in gravity sensing columella cells correlates with the onset of non-differential auxin redistribution and symmetric (non-gravitropic) LR organ growth[7]. In agreement, also the asymmetry in CK signaling declines at the onset of stage III LR, proposing that gravity-induced, differential growth at upper and lower organ flanks is generally switched off at this stage. Subsequent de-repression of *PIN3* and *PIN4* expressions in columella cells of stage III LRs does not induce alterations in GSA of LR[8], suggesting that the gravitropic machinery is similarly inactive in

these older stage III LRs. On the other hand, the onset of expression and subsequent polarization of PIN7 coincide with the stage III–IV transition, which is marked by further gravitropic bending of the respective LRs[8]. It needs to be seen how gravitropic perception and/or response is reactivated in these LRs and if CK signaling also plays a role in the incremental gravitropic responses during the stage III–IV transition of LRs.

In conclusion, our genetic screen uncovered that directional LR growth depends on opposing gravitropic and anti-gravitropic phytohormonal cues (Supplementary Fig. 8h). We conclude that CK signaling reduces growth at the upper organ side, which counteracts the gravity induced, auxin-dependent reduction in cell expansion at the lower root flank. In this way, a CK-dependent mechanism allows the root system to override the gravitropic response and radially explore its surroundings. Genetic interference with CK signaling cannot only be used to define the primary growth direction of LRs, but moreover may refract certain environmental input to root architecture. Overall, these results propose that directed interference with CK responses in LRs could be used to engineer root system depth to better suit certain habitats.

## Methods

**Plant material and growth conditions.** Seeds of *Arabidopsis thaliana* accessions were kindly provided by Magnus Nordborg and Wolfgang Busch. The following lines and constructs have been described previously: *ahk2-7* (*ahk2*), *ahk3-7* (*ahk3*)[39], *cre1-12* (*ahk4*)[40], *ahk2 ahk3*, *ahk2 ahk4*, *ahk3 ahk4*[39], *arr3*, *arr4*, *arr10-5*, *arr12-1* pARR3::GUS, pARR4::GUS, pARR10::GUS, pARR12::GUS[41], CDKB1;1 DN[42], *cdkb1;1 cdkb1;2*[43], 35S::CKX2[44], *ckx2-1*[45], *crf1-2*, *crf2-1*, *crf2-2*, *crf3-1*, *crf3-2*, RPS5a::CRF2, 35S::CRF3[24], pCRF2::GFP/GUS, pCRF3::GFP/GUS[46], pCYCB1:: GUS[47], pDR5::GFP[6], pPIN3::PIN3-GFP[48], TCSn::GFP[32], TCSn::GFP *ahk2 ahk4*[49], pUBQ10::LUC2o[15]. Seeds of *Brassica napus* L. were kindly provided by Saatzucht Donau. Seeds were surface sterilized, stratified at 4 °C for 2 days in the dark. Seedlings were grown vertically on half Murashige and Skoog medium (1/2 MS salts (Duchefa), pH 5.9, 1% sucrose, and 0.8% agar). Plants were grown under long-day (16 h light/8 h dark) conditions at 20–22 °C.

**Chemicals and treatments.** 1-N-naphthylphthalamic acid (NPA) (Sigma), 6-Benzylaminopurin (BAP) (Sigma), *trans*-Zeatin (tZ) (OlChemim), N[6]-(2-Isopentenyl)Adenine (iP) (OlChemim) were all dissolved in DMSO (Duchefa). 2-chloro-6-(3-methoxyphenyl)aminopurine (INCYDE) was synthesized and kindly donated by the Laboratory of Growth Regulators, Palacký University & Institute of Experimental Botany AS CR (Olomouc, Czech Republic) as previously described[19] and dissolved in DMSO. Treatments with NPA BAP, tZ, iP and INCYDE were all performed on 7-day-old seedlings (transferred to supplemented media).

GUS stainings were performed after 24 h and initial GSA measurements 7 days after transfer.

**Genome-wide association studies (GWAS).** To identify the genetic basis of the for the GSA of LRs we carried out a GWAS using an accelerated mixed model (AMM)[16]. The GWAS results can be viewed interactively online: https://gwas.gmi.oeaw.ac.at/#/analysis/12722/overview.

**Rhizotrons.** The basic rhizotron design was as described in ref. [15]. To adapt the rhizotrons for use in an automated rhizotron handling system (designed by Modular Science, San Francisco), several modifications were implemented. The top edge of each rhizotron sheet was beveled using a belt sander to facilitate automated watering. Two 1/16" thin aluminum hooks used for automatic handling of the rhizotron were attached on each side of the rhizotron. To reduce light exposure of the root system during growth, a 1/8" thin black acrylic rhizotron top shield was installed.

**Box and holders for rhizotrons.** Black 12" W × 18" L × 12" H boxes (Plastic-Mart) were used to grow plants in 12 rhizotrons at a time. The arrangement of 1/8"-thin black acrylic sheets of different shapes and sizes formed 12 light-proof chambers to make sure that the roots of every rhizotron were shielded from light even when one rhizotron was removed for imaging.

Rhizotron preparation was as described in ref. [15] with slight modifications required by the new rhizotron design.

**Plant growth in rhizotrons.** Two transfer pipettes (each ~2 ml) of quick-releasing fertilizer (Peter's 20–20–20) were added to each rhizotron after assembly. Assembled rhizotrons were placed into a box with water and allowed to absorb water overnight from the bottom side. Seeds containing the pUBQ10::LUC2o transgene[15] were stratified for 2 days at 4 °C in distilled water and three seeds were sown in the center of each rhizotron. Each rhizotron was equipped with a unique barcode. All rhizotrons were sprayed down with water and sealed with a transparent lid and packing tape. Plants were grown at 22/18 °C (day/night) under long-day conditions (16 h light, 8 h dark) using LED lights (Vayola, C-Series, N12 spectrum) with a light intensity of about 130 μmol m$^{-2}$ s$^{-1}$. After 2 days, the transparent lid was unsealed, rhizotrons were watered with two transferring pipettes of water, and the lid left loose for an additional day. After removing the lid, rhizotrons were watered twice per day with two transferring pipettes of water each time until 9 days after sowing.

**Plant imaging in rhizotrons.** 20 days after sowing, the automated rhizotron handling system (designed by Modular Science, San Francisco) added 50 ml of 300 μM D-luciferin (Biosynth) at the top of each rhizotron and loaded the rhizotron into a fixed stage that was controlled by a Lambda 10-3 optical filter changer (Sutter Instruments, Novato, CA) in the GLO1 imaging system[15]. 5-min exposures were taken per rhizotron side.

A shoot image was taken right after the four root images using an ids UI-359xLE-C camera with a Fujinon C-Mount 8–80 mm Varifocal lens that was installed in GLO1. Three LED strips on each side of the camera were switched on before a shoot image was taken.

**Image preparation of rhizotrons pictures.** Image preparation was similar to that in ref. [15]: four individual root images were collected: top front, bottom front, top back, and bottom back. Using an automated ImageJ macro, a composite image was generated as follows: (1) images were rotated and translated to control for small misalignments between the two cameras; (2) the top and bottom images of each side were merged; (3) the back image was flipped horizontally; (4) the front and back images were combined using the maximum values. The final images produced were 16-bit in depth and 4096 × 2048 pixels. The scale of the images was 138.6 pixels per cm.

**Hypoxia treatment.** All following treatments were performed in air-tight glass desiccators in which seedlings grown on vertical agar plates were carefully placed with the lids removed. Seedlings were exposed to a hypoxia treatment by flushing the desiccators with humidified 100% N$_2$ gas (2 l/min) for 4 h (13.00–17.00 h) in the dark to limit photosynthesis-derived oxygen production. For the controls, desiccators were flushed with humidified air. Flow rates were controlled by mass flow controllers (MASS-VIEW, Bronkhorst). At the end of the hypoxia treatment, plates were carefully removed from the desiccators, closed, and transferred back to the climate chamber. The plates remained in the climate chambers under control growth conditions for 5 days after the treatment after which they were scanned using an EPSON Scanner V300.

**DNA constructs.** The promoter region and full-length CKX2$^I$ or coding DNA sequence (CDS) were amplified by PCR (Supplementary Data 3) from genomic DNA or cDNA using Q5 high-fidelity DNA polymerase (NEB) and cloned either alone or under of the 35S promoter together with GFP and mScarlet-i into pPLV03 or pGEX5×3 using Gibson Assembly Master Mix (NEB). Subsequently, this plasmid were used for in vitro mutagenesis (Supplementary Data 3) to obtain CKX2$^M$. The resulting constructs were transformed into Col-0 and *ckx2-1* plants using the floral dipping method[50] or for transient transformation in tobacco plants.

**Activity measurement of recombinant proteins.** Recombinant proteins were expressed as GST fusion proteins and in *Escherichia coli* BL21 codon plus strain. Proteins were purified using the Sepharose beads affinity method (Glutathione Sepharose 4B; GE Healthcare).

The activity was measured using a modified end-point method previously described[51]. In brief, the samples were incubated in a reaction mixture (total volume of 600 μl) that consisted of 200 mM McIlvaine buffer (100 mM citric acid and 200 mM Na$_2$HPO$_4$) pH 7.4, 500 μM 2,6-dichlorophenol indophenol (DCPIP; Sigma) as electron acceptor and different concentrations of N[6]-(2-isopentenyl) adenine (iP; Sigma) as substrate. The volume of the enzyme sample used for the assay was adjusted based on the enzyme activity. The incubation time at 37 °C was 1 h. The enzymatic reaction was stopped after incubation by adding 300 μl of 40% trichloroacetic acid (TCA), then 200 μl 2% 4-aminophenol (Sigma) (in 6% TCA) was added and the sample was centrifuged at 20,000 × *g* for 5 min to remove protein precipitate. 200 μl supernatant was used to measure the absorption spectrum from 352 to 500 nm to determine the concentration of produced Schiff base with ε352 = 15.2 mM$^{-1}$ cm$^{-1}$ using a plate reader.

**Microscopy.** Confocal microscopy was performed using a Leica SP5 (Leica). Fluorescence signals for GFP (excitation 488 nm, emission peak 509 nm), mScarlet-i (excitation 561 nm, emission peak 607 nm), mTurquoise (excitation 434 nm, emission peak 474 nm) and propidium iodide (PI) staining (20 μl ml$^{-1}$) (excitation 569 nm, emission peak 593 nm) were detected with a ×40 or ×63 (water immersion) objective. The fluorescence signal intensity (mean gray value) of the presented

markers was quantified using the maximum projections obtained from a Z-stack series that were taken and analyzed using the Leica software LAS AF 3.1. The same region of interest (ROI) was defined for each individual seedling.

To determine meristematic cell numbers and cell size in stage II LRs, 8-day-old seedlings ($n = 10–15$) were stained with PI and microscopy was performed using confocal microscope (see above). The first two epidermal cells (adjacent to the main root) were considered for cell size measurements in stage II LR. Epidermal cell numbers were counted between the quiescent center and the first elongating cell (twice as long as wide) at the upper and lower flank of stage II LRs.

Graphpad Prism software was used to evaluate the statistical significance of the differences observed between control and genotype/treatments (one-way ANOVA).

**Gravitropic set-point angle measurements**. Plates with 14-day-old seedlings were scanned and the initial gravitropic set-point angle (iGSA) of individual LRs was measured with reference to the gravity vector[7] using Image J software. All available LR from each seedling were measured. Individual GSA values were then sorted into eight categories: 0–30°, 31–50°, 51–70°, 71–90°, 91–110°, 111–180°. Percentages of incidence were calculated for each category and graphs of GSA distribution were generated. The test of Kolmogorov–Smirnov (KS-test) was used online (http://www.physics.csbsju.edu/stats/KS-test.n.plot_form.html) to statistically evaluate the GSA data sets generated from mutants and treated seedlings in comparison to wild type and untreated controls, respectively.

**Histochemical GUS staining**. GUS histochemical staining of acetone-fixed 7-day-old seedlings containing pCycB1::GUS constructs followed a previously described method[52] using 5-Bromo-4-chloro-1*H*-indol-3-yl β-D-glucopyr-anosiduronic acid (X-Gluc, Carl Roth) as substrate. In brief: seedlings were fixed in 90% aceton for 30 min. After washing with 0.1 M Na-phosphate buffer (pH 7) seedlings were incubated for 2 h at 37 °C in the GUS staining solution (2 mM x-Gluc (dissolved in DMSO), 0.1% Trition X-100, 10 mM EDTA, 0.5 mM potassium ferrocyanide, 0.5 mM ferrycyanide, 0.1 M Na-phosphate buffer pH 7). Examination of stained seedlings and image acquisition were performed with a light microscope (Zeiss Observer D1) equipped with a DFC 300 FX camera (Zeiss). The intensity of the staining was quantified as described ref.[53] in a region of interest (ROI), which was kept constant. Graphpad Prism software was used to evaluate the statistical significance of the differences observed between control and treated groups (One-way ANOVA).

**Transient transformation and western analysis**. The *Agrobacterium tumefaciens* strain GV3101 was transformed with the respective construct and grown for 2 days at 28 °C in 5 ml Luria-Bertani (LB). The preculture was used to inoculate 25 ml LB and incubated for 4 h at 28 °C. Cells were pelleted and resuspended in 30 ml LB supplemented with 100 μM acetosyringone. After 2 h, cells were resuspended in 30 ml of 5% sucrose and infiltrated in tobacco (*Nicotiana tabacum*) leaves. Subcellular localization was examined 3 days after transformation by confocal laser scanning microscopy (see above) or leaves were ground to fine powder in liquid nitrogen and solubilized with extraction buffer (25 mM Tris, pH 7.5, 10 mM MgCl₂, 15 mM EGTA, 75 mM NaCl, 1 mM DTT, 0.1% Tween 20, with freshly added proteinase inhibitor cocktail (Roche)). Protein concentration was assessed using the Bradford method. Membranes were probed with a 1:5000 dilution of GFP antibody (#ab290, abcam) or 1:20,000 of tubulin antibody (T6074-200UL, Sigma). Goat IRDye 800CW anti-mouse (926-32210, LI-COR) or goat IRDye 800 CW anti-rabbit (926-32211, LI-COR) was used (1:20,000) as secondary. The signals were detected and quantified using the Odyssey Imagine System (LI-COR).

**RNA extraction, cDNA synthesis and quantitative PCR**. RNA extraction was done as described previously[54]. In brief: a pool of 10 LR or root tips were collected in 30 μl of 100% RNAlater (Thermo Fisher) and 500 μl of TRIzol (Sigma) was added followed by brief vortexing (2× for 2 s each) and incubating at 60 °C for 30 min. 100 μl of chloroform was added, and then, samples were vortexed briefly (2× for 2 s each) and incubated at room temperature for 3 min. After centrifugation at 12,000 × *g* for 15 min at 4 °C, the aqueous phase was transferred to a new tube. To precipitate the RNA, an equal volume of isopropanol and 1.5 μl of GlycoBlue (Thermo Fisher) were added followed by a −20 °C incubation for 15–18 h and centrifugation at >20,000 × *g* for 60 min at 4 °C. After removal of the supernatant, the pellet was washed by adding 500 μl of 75% ethanol, vortexing briefly and then centrifuged at >20,000 × *g* for 15 min at 4 °C. The 75% ethanol wash step was repeated 1×. As much ethanol as possible was removed followed by the drying of the pellet by letting the Eppendorf tube sit on ice with lid open for 10 min. Precipitated RNA was then resuspended with 5–12 μl of nuclease-free water, stored at −80 °C. cDNA synthesis was performed using SuperScript II (Thermo Fisher) and qPCR using 2x Takyon for SYBR Assay—no ROX (Eurogentec) following the manufacturer's instructions on a CFX96 Touch Real-Time PCR Detection System (Bio-Rad). Expression values were normalized to the expression of ubiquitin 5 (UBQ5) and translation initiation factor EIF4A.

**Cytokinin measurements**. Quantification of cytokinin metabolites was performed according to the method described by Svačinov et al.[55], including modifications described in ref.[56]. Briefly, root samples (20 mg FW) were extracted in 1 ml of

modified Bieleski buffer[57] together with a cocktail of stable isotope-labeled internal standards used as a reference (0.25 pmol of CK bases, ribosides, N-glucosides, and 0.5 pmol of CK O-glucosides, nucleotides per sample added). The extracts were purified using the Oasis MCX column (30 mg/1 ml, Waters) and cytokinin levels were determined using the LC–MS/MS system consisting of an ACQUITY UPLC System and a Xevo TQ-S triple quadrupole mass spectrometer (Waters). Results are presented as the average of five biological replicates ± standard deviation in pmol/g FW. Statistical examinations were made between Col-0 wild type and *ckx2-1* roots using one-way ANOVA analysis.

**Assessment of gravitropic main root growth**. We used infrared-based time lapse of gravitropic main roots (90° tilted plates) in the dark. Growth rate normalization was performed as described in ref.[58].

**Maps**. The maps were created in R using the package "rworldmap"[59].

**Description of the computer model of LR**. For the sake of simplicity, our model is composed of rectangular grid in which each box represents a single cell—a basic space discretization unit in the model. Cell walls are modeled as a linear elastic spring (connecting two adjacent vertices) that can expand and contract in order to minimize forces acting on each spring. The magnitude of force exerted by this spring is $k_x \cdot (L_{u,v} - |p_u - p_v|)$ and is positive for spring compression. The $k_x$ characterizes the stiffness of the spring and was set to 0.9 in all simulations. This force is in the direction of the spring $\frac{p_u - p_v}{|p_u - p_v|}$, $p_u$ is the position of vertex **u**, and $p_v$ is the position of neighboring vertex **v**. The total force exerted on vertex **u** located at position $p_u$ by all such springs can be written as

$$F_{\text{linear}}^u = \sum_{u \in N_u} k_x \cdot (L_{u,v} - |p_u - p_v|) \cdot \frac{p_u - p_v}{|p_u - p_v|}$$

where $N_u$ is the set of vertices adjacent to vertex **u**. The norm symbol indicates the Euclidean distance between the points.

In addition to the forces acting on a vertex due to springs, a force due to the turgor pressure inside the cell ($p_{\text{const}} = 0.05$) acts in the direction normal to each wall (***n***):

$$F_{\text{pressure}}^u = p_{\text{const}} \cdot \hat{n} \cdot |p_u - p_v|$$

This cell turgor pressure helps to maintain cell shape protecting from artificial deformations[60].

Combining the individual force components, the total force acting on a vertex **u** is the sum of forces acting on each cell wall and internal pressure inside the cells.

According to the second Newton's Law of dynamics we calculated the velocity (Vel$_u$) and position ($p_u$) of vertex **u** over time for point mass $m_u = 1$ with the following formulas:

$$\frac{\text{dVel}_u}{\text{d}t} = \frac{F_{\text{total}}^u}{m_u} - \beta \cdot \text{Vel}_u, \frac{\text{d}p_u}{\text{d}t} = \text{Vel}_u$$

where $\beta = 0.2$ is a damping constant.

Other details of a model setup and methodology as well as its successful applicability for resolving the organ bending simulations can be found in ref.[60]. The LR root model is spatially divided into two zones (root tip and cell elongation zones) along x-axis based on the threshold parameter that defines distance between individual cell center and right-most cell centers. The right-most boundary vertices were fixed in x-direction to mimic connection to the main root axis (outermost right) such that growth occurs only from the main root axis as observed experimentally. We use thresholds that controls either the length of whole elongation zone or simply the number of elongating cells only on the upper root flank or progressively from the top to the bottom to mimic the input of an asymmetric meristem in our simulations (Supplementary Figs. 7 and 9). Along y-axis cells elongate at different rates based on the linear interpolation between the minimal measured elongation rate at the bottom part of the LR (5 μm h⁻¹) to the maximal elongation rate at the top flank of the LR (15 μm h⁻¹)[7]. Furthermore, we found that linear interpolation gave the best fit to the experimental observations in terms of >2-fold change in cell size between upper and lower LR root flanks. To simulate reduced elongation, we consequently reduced the maximal rate of cell elongation on upper root flank by 10%. Cell elongation is simulated by expanding the resting length of linear springs at each growth time step (0.003). The growth time step occurs after mass-spring system reaches the transient equilibrium such that the slower growth and faster mechanics steps follow consecutive iterations[60].

$$\frac{\text{d}L_{u,v}}{\text{d}t} = L_{u,v} \cdot r(d)$$

where $r(d)$ is an linearly interpolated growth rate and $d$ is the relative distance from the bottom part of the LR such that $r_{\text{min}}(0) = 5$ μm h⁻¹ and $r_{\text{max}}(1) = 15$ μm h⁻¹.

The geometry of the model was created using a version of the VV simulator[61,62] embedded in the modeling software L-studio[63] (http://algorithmicbotany.org/lstudio). Cell mechanics and growth steps was solved using the forward Euler method. All simulations were terminated after 9 h of growth to match the experimental observations.

**Reporting summary**. Further information on research design is available in the Nature Research Reporting Summary linked to this article.

## Data availability

All data generated or analyzed during this study are included in this article and its supplementary information files. Seeds and plasmids are available from the corresponding author upon request. The GWAS data set has been deposited at https://gwas.gmi.oeaw.ac.at/#/analysis/12722/overview.

## Code availability

All information for building the model is indicated in the manuscript. Further information is available from the corresponding author upon request.

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

## Acknowledgements

We are grateful to Bruno Müller, Thomas Schmülling, Magnus Nordborg, Wolfgang Busch, Ben Scheres, Jiri Friml, Dirk Inze, Jungmook Kim, Tomas Werner, Marketa Pernisova, Eva Benkova, Joseph Kieber, and Lieven De Veylder for sharing published material; Marget Sauter, Ilka Reichardt-Gomez, Ümit Seren and Envel Kerdaffrec for helpful discussions; Jit Thacker for help with preparing the manuscript; Hana Martínková for help with phytohormone analyses; and the BOKU-VIBT Imaging Centre for access and expertize. This work was supported by the Austrian Academy of Sciences (ÖAW) (DOC fellowship to K.D.), Fulbright-Austria Marshall Plan student grant (to E.S.), Vienna Research Group (VRG) program of the Vienna Science and Technology Fund (WWTF) (to J.K-V.), the Austrian Science Fund (FWF) (P29754) (to J.K-V.), the European Research Council (ERC) (Starting Grant 639478-AuxinER) (to J.K-V.), Deutsche Forschungsgesellschaft fellowship (LI 2776/1-1) (to H.L.); and work was funded by the Ministry of Education, Youth and Sports of the Czech Republic (National Program for Sustainability I, grant no. LO1204) (to O.N.), and Programa de Atracción de Talento 2017 (Comunidad de Madrid, 2017-T1/BIO-5654 to K.W.).

## Author contributions

S.W. performed most experiments. M.R.R. initiated the project. M.S., E.S. and K.D. performed confocal microscopy. H.L., T.L.R. and J.R.D. contributed GLO-Roots data. I.P. and O.N. conducted quantification of endogenous cytokinins. S.M. and R.S. performed hypoxia experiments K.W. designed and described the dynamic computer model simulation. J.K.-V. devised and coordinated the project. S.W. and J.K.-V. wrote the manuscript. All authors saw and commented on the manuscript.

## Additional information

**Competing interests:** The authors declare no competing interests.

