## [Peer Review File · Nature Communications]

Reviewers' comments:

Reviewer #1 (Remarks to the Author):

Directional growth of lateral roots is critical for radial root expansion in soil. Lateral roots emerge from the primary root at a 90 degree angle (Stage I) and start to bend downward as they elongate (Stage II). Different from primary roots, lateral roots display a partially suppressed gravitropic response and maintain the primary GSA during the Stage III. In this study, Waidmann and coauthors present a significant new advance in our understanding of the molecular mechanisms regulating the lateral root directional growth. First, authors performed a genome-wide association study (GWAS) and found the CYTOKININ OXIDASE2 (CKX2) gene could be linked to the natural variations of radial root growth in Arabidopsis. CKX2 encodes an enzyme that affects the phytohormone cytokinin metabolism. Then, authors revealed that the extent of downward bending of lateral roots is closely correlated with the asymmetric cytokinin signaling across the lateral root organ. Authors also constructed a dynamic computational model to simulate cell elongation and cell proliferation during lateral root bending. And, genetic and pharmacological data experimentally demonstrated that cytokinin modulate both cell elongation and proliferation at the upper flank of lateral roots.

The described experiments and quantitative analysis were carefully designed and executed, data were well documented and results were thoroughly discussed. Overall this is a well written data rich, and reads easily manuscript.

Major concerns:

- 1) In Line 351, authors claimed that not only the cellular elongation, the cell proliferation is also a particular determinant of directional lateral root growth. In Fig.7f, *ckx2* showed a significant reduction of meristem length at the upper flank compared with wild type. In Fig. S7g,h, the ratio of pCYCB1;1:GUS intensity at the upper and lower sides of lateral roots was affected by BAP or INCYDE treatment. This result suggests cytokinin has different impacts on the cell division at upper and lower sides of lateral roots. Whether the cell numbers (cell proliferation) at upper and lower side are affected differentially by cytokinin?
- 2) In computational model, Fig. 7 and Fig. S7, the cell elongation rate and number of elongating cells within the elongation zone were simulated. In Fig. S7e, only the length of elongated cells (the two epidermal cells next to the main root) were compared. It will be appreciated if authors could plot the cell length within one cell file at upper side and one at the lower side of Stage II lateral roots.
- 3) In Fig. 7g and 7h, whether/how the cell proliferation and cell elongation in *CDKB1;1* DN/*cdkb1;1* *cdkb1;2*, and Roscovitine-treated lateral roots is associated with their altered GSAs?
- 4) How the quantification of fluorescent intensity (such as TCSn::GFP, CYCb1;1LGUS) was performed?

Minors:

- Fig.2 labeling of panel 'C' and 'D', should be in lowercase
Fig.3 and Fig. S4, legends, 'Scale bar, 25 μ M' should be 'm'

Reviewer #2 (Remarks to the Author):

In this manuscript, the authors postulate that natural variation at Cytokinin Oxidase 2 (CKX2) impacts the gravitropic set point angle (GSA) in Arabidopsis through conducting GWAS study, complementation test, physiological and molecular analyses. They also demonstrated that CKX2 does not directly affect auxin signaling in emerged lateral roots unlike primary roots, suggesting that a mechanism of gravitropic response in lateral root is different from that of primary root. I think their results are very interesting. And, most of experimental designs and data with appropriate statistical analyses shown in their manuscript support their findings. However, there

are some essential points for their conclusion that need to be addressed by the authors.

Major points

Comment 1. The authors concluded that cytokinin (CK)-dependent mechanism function as anti-gravitropic component in lateral roots. Their data suggest that CK-dependent mechanism is more important compared to auxin signaling in GSA formation of lateral roots. There are many data for lateral roots, but no data for primary roots in Figs. 6 and 7. I consider that both the data of the lateral and primary roots are necessary for their conclusion. For example, in Fig. 6b, data of MR has been taken only on one developmental stage. But, I wonder whether a level of CKX2 expression in each developmental stage of primary root is different or not.

Comment 2. I have a feeling of strangeness with the flow of story although the data in this paper is well organized. For example, contents of subheading "Single base-pair variation in CKX2 impacts on its in-planta activity" should be put after GWAS result because readers would be easy to understand their story. Another contents of subheading "Cytokinin signaling integrates environmental cues into angular growth of lateral roots" is difficult to connect results in other story. Therefore, I recommend them to rearrange the contents as follows.

1. Angular lateral root growth displays substantial natural variation in *Arabidopsis thaliana*
2. Genome wide association study reveals a link between cytokinin metabolism and angular growth of lateral roots
3. Single base-pair variation in CKX2 impacts on its in-planta activity
4. Cytokinin Response Factors define angular growth of lateral roots
5. CKX2 does not detectably interfere with auxin signaling in emerged lateral roots
6. Emerged lateral roots display asymmetric cytokinin signaling
7. CKX2 activity determines cellular elongation in emerged lateral roots
8. Cytokinin-dependent interference with cell division rates defines angular growth of lateral roots
9. Cytokinin signaling integrates environmental cues into angular growth of lateral roots

Comment 3. Personally, I am very interested in the relationship between oxygen deficiency and GSA formation of lateral roots. However, their hypothese of lines 177 to 182 is quite speculative, and the authors must show additional data to prove their guess. I have some questions in this regard. Why did they focus only on Swedish accessions? Did they investigate about other G allele of CKX2 in other accessions other than Swedish accessions? Is it possible to avoid mutant from hypoxia condition using its slightly shallower lateral roots compared to those of wild types shown in Fig. 4c? Not only hypoxia condition, but also ethylene treatment does not happen the same phenomena of GSA in lateral root? There is a possibility that the lateral roots have become shallow by ethylene not hypoxia condition. Did they conduct ethylene treatment other than hypoxia condition?

Comment 4. Lines 282-283: they mentioned that 'we did not observe asymmetric CK signaling in unstimulated or gravity-stimulated primary roots (Fig. S6f-g)'. On the other hand, they mentioned that 'cells on the upper and lower flanks of emerged LRs show differential elongation for about 8-9 hours.' in lines 299-300. I wonder whether asymmetric CK signaling would be occurred in unstimulated or gravity-stimulated primary roots if they take data in a longer span like lateral roots. Furthermore, data of both primary and lateral roots sampled at the same timing should be shown.

Other points

Comment 5. Lines 120-121: Do you have phenotypic data of mutant roots?

Comment 6. It is difficult to see the frequency distributions throughout the all figures. The authors should devise.

Comment 7. Fig.2d: Make D lowercase.

Comment 8. Fig. 4b: It is hard to see T with blue in the left figure.

Comment 9. Fig.7: There are two figures shown by c.

Reviewer #3 (Remarks to the Author):

In this manuscript the authors apply a range of methodologies to demonstrate how asymmetric CK signalling contributes to the determination of GSA in lateral roots. Lateral root GSA is an important determinant of RSA and is a potential target for breeding more resilient or high yielding crops. While the experimental results are extensive, well documented, original and convincing I have some concerns regarding the modeling used in the manuscript.

On the experimental part of the paper I have one major comment:

On page 9, line 334 the authors mention that the meristem size in the upper flank of CK treated roots is significantly reduced relative to the wild type. It would be good if the authors could also present data for the meristem size in the lower flank of roots. These data should be used to answer what happens under normal, asymmetric CK signalling conditions, whether meristem size decreases more on the upper than lower flank due to CK signalling asymmetry. Additionally, given the demonstrated decrease in upper flank meristem size under CK application, these data should be used to answer whether this decrease applies equally to the lower flank, or whether upper and lower flank differences become amplified.

Regarding the modeling part I have several major comments

1. The Methods section currently contains too little data. As an example, in Fig 7. at the upper and lower flank different elongation speeds have been implemented but it is unclear how elongation speeds in the regions in between the upper and lower flanks are interpolated between these two values. Additionally, boundary conditions, for example whether particular boundary points are fixed, are not described. Also, no numerical details on the timestep and spacestep of the integration method are discussed, nor whether equations are simulated until mechanical equilibrium before a further elongation is applied, nor whether model outcomes are stable against further decreasing time or spacestep.

2. In the model nearly square shaped cells are used that appear to undergo only a highly limited expansion (how much % relative to start length of cell?), whereas in real plant roots the elongation zone is characterized by rapidly elongating cells, reaching more than 10 times their original length in ~8 hours. Modeling the mechanics of this highly anisotropic growth is far from trivial and more complex models are required for this (e.g. Fozard et al., 2013) than the one used here. The authors fail to argue why their simplistic approach would be sufficient for the questions they aim to answer here.

3. In Supplementary Figure 7 simulations with reduced number of elongating cells are shown. In Figure c it appears that as cell numbers decrease mechanical or numerical instabilities arise and strange deformations at the upper left boundary of the simulated tissue occur. Importantly, these deformations appear to strongly interfere with the preferential elongation at the upper flank, and seem to play a strong role in reducing the bending.

This raises the possibility that these deformations -which seem to increase with lower cell numbers-, rather than the reduction in numbers per se reduces tissue bending. This puts into question the validity of the models outcomes, and its interpretation.

4. Simulations demonstrating how a decrease in numbers of elongating cells at only/mostly the upper side counteract bending are lacking.

5. Model output is very hard to see, e.g. meristem region is black, but so are surroundings, gradients of elongation rates are near invisible.

Reviewer #4 (Remarks to the Author):

Waidmann et al. have searched for factors regulating the gravitropic set point angle (GSA) of lateral roots in *Arabidopsis*, which is established early after emergence of lateral roots and one determinant of the radial expansion of the root system. Thus far it is known that auxin and PIN3, an auxin transporter, gave a role in setting the GSA. They identify by GWAS a G-T mutation in the cytokinin-degrading CKX2 gene of *Arabidopsis* as being causal for an altered GSA of root branches and develop the concept of cytokinin acting in lateral roots as opponent of auxin. More specifically, they show that the mutation they have identified in the CKX2 gene alters signal peptide processing of the CKX2 protein and thus alters cytokinin degradation. Cytokinin is proposed to inhibit growth at the upper flank of lateral roots and thus root bending which is supported by a variety of pharmacological assays and genetic analyses. This is interesting work reporting a new discovery, the manuscript is well written and the majority of conclusions justified. I have a number of comments on the result section.

Figure 1a. Essentially nothing can be seen in this figure, it can be omitted or transferred as a larger version to supplements.

Fig. 1b. Indicate accession on y-axis. Again the figure is too small, blue lines cannot be distinguished. These tiny figures are a general problem. It does not make much sense to try to include as much data as possible which is then difficult to study (and appreciate).

Authors should clarify in how far the GSA determines the soil volume explored by roots. It seems from Fig. 1c/Suppl. Fig. 1 that there are no big differences between the accessions as later root bending compensates for initial difference in GSA. Is that so? Which lateral roots have been measured to determine GSA?

Fig. 2. A single T-G mutation in CKX2 is found as causative for altered GSA, the G allele associated with increased GSA. Here it would be helpful to include some quantitative data in the text. For example, an impact of BA on GSA is visible in Fig. 2d, but difficult to evaluate (it seems to be low). Similarly, cytokinin receptor mutation seem to have a low impact on GSA (Fig. 2c) but is commented in the text as "Conversely, CK receptor mutants show accelerated bending .." as if a major input would have been found. The small shifts towards a bit higher or lower numbers of roots belonging to the different classes indicate just a trend. A more accurate/detailed description would make it easier to evaluate the impact of cytokinin.

After having explored the relevance of CKX2 and cytokinin (pharmacological assays) the authors move on to test the relevance of CRFs. This is a bit surprising as CRFs operate downstream of type B ARR which are the transcriptional regulators operating in the two component system mediating the CK response. Why B-type ARRs were excluded from the analysis? They would have been the obvious choice. In fact, CRF2 and CRF3 are part of a larger TF gene family which had been named initially as Cytokinin Response Factors but whether all members and to which extent they are operating in the cytokinin pathway is as yet unclear. Therefore the naming may be misleading. Currently I disagree with the global conclusion on the role of CRFs in cytokinin signaling as expressed in lines 164/165. It would be necessary to show that the *crf2 crf3* mutant GSA is insensitive to the action of cytokinin. This would be the critical test to link their action in this context to cytokinin. Surprisingly, the publication by Jeon et al. (*Plant Cell* 2018) documenting a role of CRF2 and CRF3 in lateral root formation has not been considered.

Furthermore, authors may rely also not only the gene expression data by Brady et al. but those summarized by Parizot et al. (*Plant Physiol* 153, 34-40, 2010) on gene expression in lateral root formation might provide a rich and helpful source on information to identify relevant genes.

Figure 4. Geographical distribution of *Arabidopsis* accession in Sweden is correlated with the occurrence of the T and G alleles of CKX2, which is an interesting finding. Unfortunately Fig. 4b is hardly readable, dots are difficult to identify. It is a bit surprising that authors argue with a longer snow coverage and hypoxia problems associated with it. Under the snow there is no *Arabidopsis* growing or am I missing a point here? Could it be, in contrast, that the soil in Northern regions is still frozen below a certain depth and more shallow roots would avoid this uncomfortable region? Nevertheless, the impact of hypoxia on GSA is interesting.

Fig. S3a-e The cytokinin content in *ckx2-1* mutant of *Arabidopsis* is mostly decreased as compared to the control while the opposite would be expected and statements in the text seem to indicate just these opposite changes. In any case it is clearly wrong to state in the text that as expected cytokinin levels were increased in the *ckx2* mutant. Possibly the resolution of these measurements was too low as whole seedlings were analyzed. A more detailed and precise consideration of the different cytokinin metabolites is needed, otherwise it is questionable whether any relevance be attached to this data.

Figure 5. Authors developed a radiometric assay to measure CKX2 protein stability and demonstrate that SP processing is required for full activity of the protein. Complementation of the *ckx2* mutant is only possible with WT allele but not with G allele. This is a good point.

Fig. 6a shows that expression pattern of CKX2 is uniform (as is that of CRF2/3). It remains enigmatic how the differential input of CKX2 and CRFs on root growth is realized when the expression of these genes does not show a preference of the upper or lower side of the root. This point is brought up by the authors in the discussion. An idea how that might be realized is welcome.

Figure 6. The investigation of impact on auxin yields essentially negative results. I think this part could be positioned after evaluation the influence of CK signaling on cell growth and division.

The number of cells on the upper and lower side of lateral roots of both G and T lines should be compared to provide support for the hypothesis that an altered cell number contribute to altered GSA. This would complement data on cell length in Fig. 7e.

Minor points.

"D" in Fig. 2d should be "d", and at other occasions, for example Fig. S3A-E in the text is Fig. S3a-e)

Fig. 6, title, asymmetric is asymmetrical

Line 308, spring is string

Reviewers' comments:

Reviewer #1 (Remarks to the Author):

Directional growth of lateral roots is critical for radial root expansion in soil. Lateral roots emerge from the primary root at a 90 degree angle (Stage I) and start to bend downward as they elongate (Stage II). Different from primary roots, lateral roots display a partially suppressed gravitropic response and maintain the primary GSA during the Stage III. In this study, Waidmann and coauthors present a significant new advance in our understanding of the molecular mechanisms regulating the lateral root directional growth. First, authors performed a genome-wide association study (GWAS) and found the CYTOKININ OXIDASE2 (CKX2) gene could be linked to the natural variations of radial root growth in Arabidopsis. CKX2 encodes an enzyme that affects the phytohormone cytokinin metabolism. Then, authors revealed that the extent of downward bending of lateral roots is closely correlated with the asymmetric cytokinin signaling across the lateral root organ. Authors also constructed a dynamic computational model to simulate cell elongation and cell proliferation during lateral root bending. And, genetic and pharmacological data experimentally demonstrated that cytokinin modulate both cell elongation and proliferation at the upper flank of lateral roots.

The described experiments and quantitative analysis were carefully designed and executed, data were well documented and results were thoroughly discussed. Overall this is a well written data rich, and reads easily manuscript.

> Thank you for your positive and encouraging words.

Major concerns:

1) In Line 351, authors claimed that not only the cellular elongation, the cell proliferation is also a particular determinant of directional lateral root growth. In Fig.7f, *ckx2* showed a significant reduction of meristem length at the upper flank compared with wild type. In Fig. S7g,h, the ratio of pCYCB1;1:GUS intensity at the upper and lower sides of lateral roots was affected by BAP or INCYDE treatment. This result suggests cytokinin has different impacts on the cell division at upper and lower sides of lateral roots. Whether the cell numbers (cell proliferation) at upper and lower side are affected differentially by cytokinin?

> Thank you for raising this question. As requested, we counted the cell number in Col-0 and *ckx2-1* mutants. Compared to wild type stage II lateral roots, the *ckx2-1* mutants displayed asymmetric meristems, displaying more and less cells in the lower and upper flank, respectively. Accordingly, we conclude that there is a weak, but statistically significant impact of cytokinin on differential cell proliferation. We added this data to the revised version of the manuscript (Figure 7f and 7h).

2) In computational model, Fig. 7 and Fig. S7, the cell elongation rate and number of elongating cells within the elongation zone were simulated. In Fig. S7e, only the length of elongated cells (the two epidermal cells next to the main root) were compared. It will be appreciated if authors could plot the cell length within one cell file at upper side and one at the lower side of Stage II lateral roots.

> As requested, we provide additional model predictions displaying differences in average cell size (cell area) between upper and lower LR flanks (see Supplemental Fig. 7b, d, f).

3) In Fig. 7g and 7h, whether/how the cell proliferation and cell elongation in CDKB1;1 DN/cdkb1;1 cdkb1;2, and Roscovitine-treated lateral roots is associated with their altered GSAs?

> Thank you for raising this point. We assessed the number of meristematic cells in *Col-0*, *CDKB1;1 DN*, and *cdkb1;1 cdkb1;2* and found that there are not only less meristematic cells in the mutants, but that they also display asymmetric meristems, displaying significantly more cell divisions in the lower flanks when compared to the upper LR flank. We added this data to the Figure 7h.

4) How the quantification of fluorescent intensity (such as TCSn::GFP, CYCb1;1LGUS) was performed?

> We added a more detailed description of signal quantification in the Material and Methods section. Moreover, we highlighted the regions of interests in some representative images (see especially updated Figure 6).

Minors:

Fig.2 labeling of panel 'C' and 'D', should be in lowercase

Fig.3 and Fig. S4, legends, 'Scale bar, 25 μ M' should be 'm'

> Thank you for highlighting these typos. We corrected the labelling mistakes.

Reviewer #2 (Remarks to the Author):

In this manuscript, the authors postulate that natural variation at Cytokinin Oxidase 2 (CKX2) impacts the gravitropic set point angle (GSA) in Arabidopsis through conducting GWAS study, complementation test, physiological and molecular analyses. They also demonstrated that CKX2 does not directly affect auxin signaling in emerged lateral roots unlike primary roots, suggesting that a mechanism of gravitropic response in lateral root is different from that of primary root. I think their results are very interesting. And, most of experimental designs and data with appropriate statistical analyses shown in their manuscript support their findings. However, there are some essential points for their conclusion that need to be addressed by the authors.

> Thank you for your encouraging and helpful comments.

Major points

Comment 1. The authors concluded that cytokinin (CK)-dependent mechanism function as anti-gravitropic component in lateral roots. Their data suggest that CK-dependent mechanism is more important compared to auxin signaling in GSA formation of lateral roots. There are many data for lateral roots, but no data for primary roots in Figs. 6 and 7. I consider that both the data of the lateral and primary roots are necessary for their conclusion. For example, in Fig. 6b, data of MR has

been taken only on one developmental stage. But, I wonder whether a level of CKX2 expression in each developmental stage of primary root is different or not.

> Most of our data on main roots is in the Supplemental data section. We show that CRF2 and CRF3 are stronger expressed in lateral roots and that CKX2 is largely absent in the main root. Besides this, we show that the cytokinin reporter TCSn is not asymmetric in gravity stimulated main roots, suggesting that the cytokinin-dependent asymmetric interference with cell expansion and division is specific for lateral roots. In agreement, cell cycle mutants display reduced bending of lateral roots, but are not defective in main root gravitropism.

In response to the reviewer, we checked the expression of CKX2 also in younger primary roots (2d, 4d, 6d) and did not detect CKX2 expression in primary roots (see panel below). In agreement, gravitropic main root growth of *ckx2* mutants is largely not distinguishable from wild type. We added this data in the revised version of the manuscript (Supplemental Figure 6b).

Comment 2. I have a feeling of strangeness with the flow of story although the data in this paper is well organized. For example, contents of subheading “Single base-pair variation in CKX2 impacts on its in-planta activity” should be put after GWAS result because readers would be easy to understand their story. Another contents of subheading “Cytokinin signaling integrates environmental cues into angular growth of lateral roots” is difficult to connect results in other story. Therefore, I recommend them to rearrange the contents as follows.

1. Angular lateral root growth displays substantial natural variation in *Arabidopsis thaliana*
2. Genome wide association study reveals a link between cytokinin metabolism and angular growth of lateral roots
3. Single base-pair variation in CKX2 impacts on its in-planta activity
4. Cytokinin Response Factors define angular growth of lateral roots
5. CKX2 does not detectably interfere with auxin signaling in emerged lateral roots
6. Emerged lateral roots display asymmetric cytokinin signaling
7. CKX2 activity determines cellular elongation in emerged lateral roots
8. Cytokinin-dependent interference with cell division rates defines angular growth of lateral roots
9. Cytokinin signaling integrates environmental cues into angular growth of lateral

roots

> We appreciate your suggestions and your help to further improve our manuscript. We discussed your editorial suggestions among the co-authors. Our data on hypoxia is a non-essential side line in our story and we would not like to complete the manuscript with this set of data, because we feel it would give this aspect too much weight. Hence, we preferred to keep our flow of the story.

Comment 3. Personally, I am very interested in the relationship between oxygen deficiency and GSA formation of lateral roots. However, their hypothesis of lines 177 to 182 is quite speculative, and the authors must show additional data to prove their guess.

> We agree that our results do not establish the adaptive significance of shallower roots in hypoxic soils. In the result section we only conclude that our data suggests that this cytokinin-dependent mechanism has developmental importance for integrating environmental cues. We speculate only in the discussion section that this could be an adaptive, avoidance mechanism to position roots closer to the soil surface. Addressing the latter is however out of the scope of this manuscript and we clearly state in the revised version of the manuscript that additional experiments are needed to validate these assumptions.

I have some questions in this regard. Why did they focus only on Swedish accessions? Did they investigate about other G allele of CKX2 in other accessions other than Swedish accessions?

> We focused on the Swedish accessions, because here the minor G allele is the most prevalent allele. We also noted other interesting aspects in allele distributions. For example: G alleles in the south of Spain do not display the cckx2 loss of function phenotype, suggesting second site mutations that suppress higher lateral root angles. We however did not yet follow up on this interesting link.

Is it possible to avoid mutant from hypoxia condition using its slightly shallower lateral roots compared to those of wild types shown in Fig. 4c?

> We exposed a very mild hypoxia treatment for only 4 hours, which was sufficient to mildly but statistically significantly alter the root system architecture. While hypoxia avoidance is indeed an interesting question, this set of data is rather a side story of our manuscript and not the main scope of this work. Additional work would be indeed needed to assess whether GSA angles close to 90° are beneficial under hypoxia conditions.

Not only hypoxia condition, but also ethylene treatment does not happen the same phenomena of GSA in lateral root? There is a possibility that the lateral roots have become shallow by ethylene not hypoxia condition. Did they conduct ethylene treatment other than hypoxia condition?

> Ethylene is indeed also an important stress signal accumulating in flooded plants. However, in the experiments conducted here, seedlings were flushed with 100% N₂ gas (See Methods) leading to steep decline in oxygen levels (under an hour)

approaching anoxia. These conditions are unlikely to enhance ethylene production (since the terminal step of ethylene biosynthesis requires oxygen). Moreover, when we transferred seedlings to plates containing 1 μ M of the ethylene precursor ACC we did not see any statistically significant changes in the GSA distribution of Col-0 wild-type plants (see panel below).

Comment 4. Lines 282-283: they mentioned that ‘we did not observe asymmetric CK signaling in unstimulated or gravity-stimulated primary roots (Fig. S6f-g)’. On the other hand, they mentioned that ‘cells on the upper and lower flanks of emerged LRs show differential elongation for about 8-9 hours.’ in lines 299-300. I wonder whether asymmetric CK signaling would be occurred in unstimulated or gravity-stimulated primary roots if they take data in a longer span like lateral roots. Furthermore, data of both primary and lateral roots sampled at the same timing should be shown.

> Yes, it is correct that the whole processes of lateral root growth from emerging to end of stage III takes around 8-9 hours. However, we focused in most of our analysis on very early stage II lateral roots. At the onset of stage II (when lateral roots become competent to respond to gravity) we detect both asymmetric auxin and asymmetric cytokinin signalling. This suggests that the response is rather immediate or at least within hours. Based on your comments, we are now displaying the TCSn reporter in main roots after 1, 2, and 3 hours of gravistimulation. We did not detect any asymmetry. In contrast, we also show that DR5 reporter was asymmetric already after 1 hour of gravistimulation. We, hence, conclude that cytokinin signalling dynamics in lateral and main roots are distinct.

Other points

Comment 5. Lines 120-121: Do you have phenotypic data of mutant roots?

> Phenotypic data of *ckx2-1* are displayed in Figure 2G.

Comment 6. It is difficult to see the frequency distributions throughout the all figures. The authors should devise.

> This manuscript is very dense with data. To improve readability, we added an additional Supplemental file (excel sheet) with all frequency distributions.

Comment 7. Fig.2d: Make D lowercase.

> Amended.

Comment 8. Fig. 4b: It is hard to see T with blue in the left figure.

> We improved the readability of the figure in the revised version of the manuscript.

Comment 9. Fig.7: There are two figures shown by c.

> We describe now left and right panel in the revised version of the manuscript.

Reviewer #3 (Remarks to the Author):

In this manuscript the authors apply a range of methodologies to demonstrate how asymmetric CK signalling contributes to the determination of GSA in lateral roots. Lateral root GSA is an important determinant of RSA and is a potential target for breeding more resilient or high yielding crops. While the experimental results are extensive, well documented, original and convincing I have some concerns regarding the modeling used in the manuscript.

> We thank you for your encouraging words on our work.

On the experimental part of the paper I have one major comment:

On page 9, line 334 the authors mention that the meristem size in the upper flank of CK treated roots is significantly reduced relative to the wild type. It would be good if the authors could also present data for the meristem size in the lower flank of roots. These data should be used to answer what happens under normal, asymmetric CK signalling conditions, whether meristem size decreases more on the upper than lower flank due to CK signalling asymmetry. Additionally, given the demonstrated decrease in upper flank meristem size under CK application, these data should be used to answer whether this decrease applies equally to the lower flank, or whether upper and lower flank differences become amplified.

> Thank you for the suggestion. In some experiments we see a tendency of asymmetric meristems in the wild type, but it lacks statistical significance. We also quantified meristematic cell numbers in *ckx2* mutants, which reproducibly show statistically significant asymmetric meristems, displaying less cells at the upper and more cells at the lower lateral root flank. However, the impact on cell proliferation is relatively weak, suggesting that only a few cells differ at the flanks. It could indeed be that mild impact on the upper (high cytokinin) and lower (low cytokinin) flanks get amplified. We added the new data to the Figure 7f.

Regarding the modeling part I have several major comments

1. The Methods section currently contains too little data. As an example, in Fig 7. at the upper and lower flank different elongation speeds have been implemented but it is unclear how elongation speeds in the regions in between the upper and lower flanks

are interpolated between these two values. Additionally, boundary conditions, for example whether particular boundary points are fixed, are not described. Also, no numerical details on the timestep and spacestep of the integration method are discussed, nor whether equations are simulated until mechanical equilibrium before a further elongation is applied, nor whether model outcomes are stable against further decreasing time or spacestep.

> We thank Reviewer for raising this issue and for the help in improving the description of the model. In the revised computational methods section, we provide the requested explanation of conditions used, such as boundary conditions, steps of growth and mechanics and description of the mass-spring system. We have used Euler methods for integration with time step of 0.05 or less to assure a stable outcome. Typically, increasing numerical steps destabilize explicit solvers while decreasing steps results in stable solutions. As mentioned in the revised computational methods section, the cell growth is introduced at transient mechanical equilibrium similarly to that described in (Žádníková & Wabnik et al, 2016).

2. In the model nearly square shaped cells are used that appear to undergo only a highly limited expansion (how much % relative to start length of cell?), whereas in real plant roots the elongation zone is characterized by rapidly elongating cells, reaching more than 10 times their original length in ~8 hours. Modeling the mechanics of this highly anisotropic growth is far from trivial and more complex models are required for this (e.g. Fozard et al., 2013) than the one used here. The authors fail to argue why their simplistic approach would be sufficient for the questions they aim to answer here.

> In the revised manuscript, we provide updated model simulations and visualisations to illustrate that GSA angle of LR depends in part on the relative fold difference between elongation of cells on the top versus bottom root flanks (Fig. 7a-d, Supplementary Fig. 7 a-f). We experimentally observed ~3-fold difference in cell size on the opposing sides of the root (at max growth rates of 15m/h) (Rosquete et al. 2013), which is very different to the faster growth rate seen in the main root. Notably, the reviewer's comment on the 10 times increase in cell size relates rather to the main root dynamics. The data integrated in our model was sufficient to predict > 2-fold cell size differences on opposing root flanks (Supplementary Fig. 7) that is in fair agreement with experimental measures. We previously assumed that only cellular elongation drives angular growth of lateral roots (Rosquete et al. 2013). This simplified mathematical approach allowed us to test whether the observed parameters can recapitulate lateral root angles and in fact partially questioned our previous assumptions on the overall importance of cellular elongation. This guided us to look closer at meristematic activity and our data suggests that the control of cell proliferation is also an important determinant of angular lateral root growth.

On the other hand, the aspects on mechanical constrains in lateral roots are indeed highly speculative. Considering that we observe only weak meristematic asymmetries in *ckx2* mutant roots, we assume only a few cells less at the upper flank. Minor changes in cell number also do not induce major mechanical constrains in the updated model. This second modelling part is not essential for the manuscript and we, therefore, used this part of the modelling to discuss certain aspects in the discussion part only. We, however, believe that our simplified approach suffices to propose new predictions that

can be further tested in upcoming studies. We clearly state the speculative nature of these assumptions and the need to experimentally validate them. We have applied a similar modelling approach to slightly more complex mechanical problem of apical hook bending (bending angles of 90° as compare to 28° observed here), which actually led to verifiable predictions (Žádňíková & Wabnik et al, 2016). Nevertheless, we agree that the statements here on lateral root mechanics is highly speculative.

3. In Supplementary Figure 7 simulations with reduced number of elongating cells are shown. In Figure c it appears that as cell numbers decrease mechanical or numerical instabilities arise and strange deformations at the upper left boundary of the simulated tissue occur. Importantly, these deformations appear to strongly interfere with the preferential elongation at the upper flank, and seem to play a strong role in reducing the bending.

This raises the possibility that these deformations -which seem to increase with lower cell numbers-, rather than the reduction in numbers per se reduces tissue bending. This puts into question the validity of the models outcomes, and its interpretation.

> We appreciated Reviewer 3 concerns regarding tissue deformations. To solve these particular issues, we have found two plausible improvements to our model. We increased the internal turgor pressure in all our simulations and decrease error tolerance for simulating mass-spring mechanics step (see Computer model details). Next, we demonstrate that our predictions are robust to this modification as no further tissue deformations were observed (Fig. 7a-c and Supplementary Fig. 7a-f). Moreover, our data indicates that during the course of stage II lateral root development (8-9 hours) only a very small asymmetry in cell proliferation is observed. Hence, we assume that only a difference of a few (e.g. 2-3) cells would be realistic, which also did not display strong alterations in tissue layout.

4. Simulations demonstrating how a decrease in numbers of elongating cells at only/mostly the upper side counteract bending are lacking.

> Simulations of reduced cell number on the top flanks are now provided in Supplementary Fig. 7c-f of the revised version of the manuscript.

5. Model output is very hard to see, e.g. meristem region is black, but so are surroundings, gradients of elongation rates are near invisible.

> We have generally improved the visualization of computer model predictions and include additional analysis on cell size versus GSA angle.

Reviewer #4 (Remarks to the Author):

Waidmann et al. have searched for factors regulating the gravitropic set point angle (GSA) of lateral roots in Arabidopsis, which is established early after emergence of lateral roots and one determinant of the radial expansion of the root system. Thus far it is known that auxin and PIN3, an auxin transporter, gave a role in setting the GSA. They identify by GWAS a G-T mutation in the cytokinin-degrading CKX2 gene of Arabidopsis as being causal for an altered GSA of root branches and develop the

concept of cytokinin acting in lateral roots as opponent of auxin. More specifically, they show that the mutation they have identified in the CKX2 gene alters signal peptide processing of the CKX2 protein and thus alters cytokinin degradation. Cytokinin is proposed to inhibit growth at the upper flank of lateral roots and thus root bending which is supported by a variety of pharmacological assays and genetic analyses. This is interesting work reporting a new discovery, the manuscript is well written and the majority of conclusions justified. I have a number of comments on the result section.

> Thank you for your positive assessment of our work.

Figure 1a. Essentially nothing can be seen in this figure, it can be omitted or transferred as a larger version to supplements.

> We transferred a bigger version of the map to the Supplemental Figure 1a.

Fig. 1b. Indicate accession on y-axis. Again the figure is too small, blue lines cannot be distinguished. These tiny figures are a general problem. It does not make much sense to try to include as much data as possible which is then difficult to study (and appreciate).

> We increased the overall size of this figure panel and made the blue lines thicker.

Authors should clarify in how far the GSA determines the soil volume explored by roots. It seems from Fig. 1c/Suppl. Fig. 1 that there are no big differences between the accessions as later root bending compensates for initial difference in GSA. Is that so? Which lateral roots have been measured to determine GSA?

> We measured all emerged lateral roots of a respective seedling. We added this information to the Material and Methods section. Both our rhizotron layout and the *in vitro* vials were too small to assess the radial expansion of fully-grown root systems, because lateral roots would eventually touch the rims. Therefore, we cannot re-evaluate our current data to address this question. However, radial expansion of lateral root depends on the primary GSA, but also on how long the respective lateral root maintains this particular GSA. The subsequent “plateau exit response” correlates with the de-repression of PIN7 (see Rosquete et al., 2019) and is hence an independent trait. It is currently unknown how this “plateau” trait is linked to cytokinin.

Fig. 2. A single T-G mutation in CKX2 is found as causative for altered GSA, the G allele associated with increased GSA. Here it would be helpful to include some quantitative data in the text. For example, an impact of BA on GSA is visible in Fig. 2d, but difficult to evaluate (it seems to be low). Similarly, cytokinin receptor mutation seem to have a low impact on GSA (Fig. 2c) but is commented in the text as “Conversely, CK receptor mutants show accelerated bending ..” as if a major input would have been found. The small shifts towards a bit higher or lower numbers of roots belonging to the different classes indicate just a trend. A more accurate/detailed description would make it easier to evaluate the impact of cytokinin.

> All the statistically significant changes are also visible by eye, but indeed sometimes hard to appreciate by looking at a single seedling only. Even though we tried to display representative images, a single image cannot capture the variation of the phenotypes

and, hence, we feel the graphs displaying the angle distributions are very informative. Indeed, the phenotypes of the CK receptor mutants were statistically significant, but comparably weaker when compared to the treatment with CKs/ incyde, the *ckx2* mutant or the CKX2 overexpressors. As suggested, we added a more quantitative descriptions in the main text and also discussed potential redundancy in the pathway.

After having explored the relevance of CKX2 and cytokinin (pharmacological assays) the authors move on to test the relevance of CRFs. This is a bit surprising as CRFs operate downstream of type B ARRs which are the transcriptional regulators operating in the two component system mediating the CK response. Why B-type ARRs were excluded from the analysis? They would have been the obvious choice. In fact, CRF2 and CRF3 are part of a larger TF gene family which had been named initially as Cytokinin Response Factors but whether all members and to which extent they are operating in the cytokinin pathway is as yet unclear. Therefore the naming may be misleading. Currently I disagree with the global conclusion on the role of CRFs in cytokinin signaling as expressed in lines 164/165. It would be necessary to show that the *crf2 crf3* mutant GSA is insensitive to the action of cytokinin. This would be the critical test to link their action in this context to cytokinin. Surprisingly, the publication by Jeon et al. (Plant Cell 2018) documenting a role of CRF2 and CRF3 in lateral root formation has not been considered.

Indeed, it has been shown that the expression of CRF2 and CRF5 is dependent on the B-type ARRs (Rashotte et al. 2006) and ARR1 has shown to directly regulate CRF2 expression by binding to the CRF2 promoter (Jeon et al. 2016). Other B-type ARRs bind to a very similar nucleotide sequence motif and might also influence CRF2 expression (Kim et al. 2016). Based on the published lateral root expression data (from Brady et al.) ARR11 and ARR12 are most strongly expressed in lateral roots (Supplementary Figure 4b). But we could not detect expression of ARR11 and ARR12 in stages I-III lateral roots using the respective reporter lines (Supplementary Fig. 4c). Additionally, the GSA of lateral root for *arr10* and *arr12* mutant plants were similar to those of wild-type plants (Supplementary Fig. 4d). Our data does not claim that A- or B-type ARRs are not implied in the LR response, but we did not identify the respective genes yet. We updated the manuscript accordingly. In the revised version of the manuscript, we also show that *crf2* and *crf3* single mutant lateral roots are partially resistant to the action of cytokinin (Supplementary Fig. 4i).

Furthermore, authors may rely also not only the gene expression data by Brady et al. but those summarized by Parizot et al. (Plant Physiol 153, 34-40, 2010) on gene expression in lateral root formation might provide a rich and helpful source on information to identify relevant genes.

> The data from Parizot et al. provide information on gene expression in lateral root primordia (during the formation phase). Here, we are mainly interested in later time points of lateral root growth. The dataset from Brady et al. on lateral roots was sufficient to identify the role of CRF2 and CRF3 in angular growth of laterals, but failed to identify the respective ARRs. In upcoming projects, we will establish stage dependent RNA sequencing of lateral roots, which will likely identify the missing ARRs. However, this additional insight is beyond the scope of this already data rich manuscript.

Figure 4. Geographical distribution of Arabidopsis accession in Sweden is correlated with the occurrence of the T and G alleles of CKX2, which is an interesting finding. Unfortunately Fig. 4b is hardly readable, dots are difficult to identify. It is a bit surprising that authors argue with a longer snow coverage and hypoxia problems associated with it. Under the snow there is no Arabidopsis growing or am I missing a point here? Could it be, in contrast, that the soil in Northern regions is still frozen below a certain depth and more shallow roots would avoid this uncomfortable region? Nevertheless, the impact of hypoxia on GSA is interesting.

> We simplified the Figure 4b and improved its readability.

In Northern Sweden the Arabidopsis plants grow until fully established leaf rosettes. Afterwards, these plants remain under snow until next spring when they will flower. Snow coverage and rapid snow melt in spring can lead to hypoxic, waterlogged soils, a situation likely to be faced by Northern Swedish accessions. Although speculative at this point, the hypoxia response of these accessions could thus be an adaptive avoidance response to direct roots to more aerated soil regions closer to the surface. Other environmental inputs on root system architecture could be indeed envisioned.

Fig. S3a-e The cytokinin content in *ckx2-1* mutant of Arabidopsis is mostly decreased as compared to the control while the opposite would be expected and statements in the text seem to indicate just these opposite changes. In any case it is clearly wrong to state in the text that as expected cytokinin levels were increased in the *ckx2* mutant. Possibly the resolution of these measurements was too low as whole seedlings were analyzed. A more detailed and precise consideration of the different cytokinin metabolites is needed, otherwise it is questionable whether any relevance be attached to this data.

> Our data shows that cytokinin metabolism is altered in *ckx2* mutants. iP-types are the major substrates for CKXs (Galuszka et al., 2007) and our analysis shows that iP-types show upregulation in *ckx2* mutants. It is likely that the downregulation of other cytokinin types is a compensation mechanism. In agreement, cytokinin reporter TCSn is upregulated in *ckx2* mutant lateral roots, suggesting that cytokinin signalling is elevated in stage II lateral roots.

We analysed only root tissue in our work, however it might be necessary to analyse only stage II lateral roots (or even of different stages) to dissect primary from secondary effects. However, this is due to the low amount of material technically not feasible at the moment. In the revised version, we clearly point out that iP-types are the preferential substrate and that they were upregulated in *ckx2* mutant roots, whereas others showed downregulation. We also improved the discussion on this matter and also highlighted in the Material and Methods section that we used only root tissues for the analysis.

Figure 5. Authors developed a ratiometric assay to measure CKX2 protein stability and demonstrate that SP processing is required for full activity of the protein. Complementation of the *ckx2* mutant is only possible with WT allele but not with G allele. This is a good point.

> Thank you.

Fig. 6a shows that expression pattern of CKX2 is uniform (as is that of CRF2/3). It

remains enigmatic how the differential input of CKX2 and CRFs on root growth is realized when the expression of these genes does not show a preference of the upper or lower side of the root. This point is brought up by the authors in the discussion. An idea how that might be realized is welcome.

> We have to assume that additional molecular events or modifications lead to asymmetric cytokinin distribution or activities of these proteins. Besides, defined intercellular distribution of cytokinin could take place. Our future research will elaborate on these possibilities.

Figure 6. The investigation of impact on auxin yields essentially negative results. I think this part could be positioned after evaluation the influence of CK signaling on cell growth and division.

> Based on the literature, it was rather to be expected that cytokinin would impact on PIN3 in lateral roots (as it does in main and lateral root primordia). Hence, we first performed and described these (expected) auxin-related experiments. Changing the order would be possible, but is certainly a matter of taste.

The number of cells on the upper and lower side of lateral roots of both G and T lines should be compared to provide support for the hypothesis that an altered cell number contribute to altered GSA. This would complement data on cell length in Fig. 7e.

> That is a very good point. We counted the cells in the upper and lower side in Col-0 and *ckx2-1* and found a statistically significant meristematic asymmetry in the *ckx2-1* mutant (lower number of meristematic cells at the upper compared to the lower flank). We added this data to the Figure 7f and h.

Minor points.

“D” in Fig. 2d should be “d”, and at other occasions, for example Fig. S3A-E in the text is Fig. S3a-e)

Fig. 6, title, asymmetric is asymmetrical

Line 308, spring is string

> Thank you, we corrected all these typos.

REVIEWERS' COMMENTS:

Reviewer #1 (Remarks to the Author):

The authors have addressed satisfactorily all my concerns.

Some minors:

(1) In Abstract: Since the interaction between auxin and cytokinin has not been intensively investigated in this manuscript, authors should tone down this statement 'Our interdisciplinary approach revealed that two phytohormonal cues at opposite organ flanks counterbalance each other's negative impact on growth'.

(2) In Discussion: Although this manuscript focus on the role of cytokinin in setup the first GSA, considering the increasing CKX2 expression in Stage II/III lateral roots, the possible role of cytokinin and its interaction with auxin in the subsequent stages (i.e. Stage III) should be discussed at least. By the way, an important reference about the spatial regulation PINs in lateral roots during gravitropism is missing (Wang et al. 2015 Nature Communications).

(3) Figure 7: The figure title needs to be changed, for example, 'interference with cell cycle' is misleading. In the legend, it is '9 h' vs the '8 h' in Fig. 7b, white arrow heads are missing in the figure. Authors showed that the cell numbers in *cdkb1;1 1;2* or *CDKB1;1 DN* line, are much lower than that in wild-type (Fig. 7h). Whether *cdkb1* mutant/DN line exhibited a difference in the cell length, like the data shown in Fig. 7e?

Reviewer #2 (Remarks to the Author):

In the revised manuscript and response letter, the authors addressed most of my comments. I only request the authors to reconsider the position of "Cytokinin signaling integrates environmental cues into angular growth of lateral roots". This content should be moved to last order or omitted because readers would be confused about the position of this story between sections of GWAS and sections of molecular analysis of CKX2. They also mentioned that "Our data on hypoxia is a non-essential side line in our story and we would not like to complete the manuscript with this set of data" in their response letter.

Reviewer #3 (Remarks to the Author):

I am satisfied with the responses and changes made by the authors.

I have a few remaining comments:

In the simulations in which the number of elongating cells at the upper side of the tissue is varied it seems that in all other tissue layers this number is kept constant. What would happen if instead there was a smooth interpolation (resulting in a diagonal line) to gradually increase the number of elongating cells until the bottom side of the tissue is reached. This seems a more natural approach.

Typo

page 31 turgor pleasure -> turgor pressure ;)

page 32 to simulated -> to simulate

REVIEWERS' COMMENTS:

Reviewer #1 (Remarks to the Author):

The authors have addressed satisfactorily all my concerns.

Some minors:

(1) In Abstract: Since the interaction between auxin and cytokinin has not been intensively investigated in this manuscript, authors should tone down this statement 'Our interdisciplinary approach revealed that two phytohormonal cues at opposite organ flanks counterbalance each other's negative impact on growth'.

> We tuned down the statement as requested:

Our interdisciplinary approach "proposes" that two phytohormonal cues at opposite organ flanks counterbalance each other's negative impact on growth

(2) In Discussion: Although this manuscript focus on the role of cytokinin in setup the first GSA, considering the increasing CKX2 expression in Stage II/III lateral roots, the possible role of cytokinin and its interaction with auxin in the subsequent stages (i.e. Stage III) should be discussed at least.

> We added some discussion on the transition between stage III and IV and raised the questioned whether cytokinin signalling contributes.

By the way, an important reference about the spatial regulation PINs in lateral roots during gravitropism is missing (Wang et al. 2015 Nature Communications).

> We included a statement on transcriptional regulation of PINs in lateral roots.

(3) Figure 7: The figure title needs to be changed, for example, 'interference with cell cycle' is misleading. In the legend, it is '9 h' vs the '8 h' in Fig. 7b, white arrow heads are missing in the figure. Authors showed that the cell numbers in *cdkb1;1 1;2* or *CDKB1;1 DN* line, are much lower than that in wild-type (Fig. 7h). Whether *cdkb1* mutant/DN line exhibited a difference in the cell length, like the data shown in Fig. 7e?

> We updated the figure and figure title as suggested.

Reviewer #2 (Remarks to the Author):

In the revised manuscript and response letter, the authors addressed most of my comments. I only request the authors to reconsider the position of "Cytokinin signaling integrates environmental cues into angular growth of lateral roots". This content should be moved to last order or omitted because readers would be confused about the position of this story between sections of GWAS and sections of molecular analysis of CKX2. They also mentioned that "Our data on hypoxia is a non-essential side line in our story and we would not like to complete the manuscript with this set of data" in their response letter.

> As already discussed in the previous revision, we indeed would not like to finish the manuscript with this side line. We prefer to finish the manuscript with the molecular analysis of CKX2, because we also elaborate on this data most extensively in the subsequent discussion section. We appreciate the help to improve the readability of our manuscript, but

the decision here comes down to a personal taste.

Reviewer #3 (Remarks to the Author):

I am satisfied with the responses and changes made by the authors.

I have a few remaining comments:

In the simulations in which the number of elongating cells at the upper side of the tissue is varied it seems that in all other tissue layers this number is kept constant. What would happen if instead there was a smooth interpolation (resulting in a diagonal line) to gradually increase the number of elongating cells until the bottom side of the tissue is reached. This seems a more natural approach.

> We included an additional simulation, covering the raised question.

Typo

page 31 turgor pleasure -> turgor pressure ;)

page 32 to simulated -> to simulate

> amended.